# Light programmable micro/nanomotors with optically tunable in-phase electric polarization

Zexi Liang [1], Daniel Teal [2] & Donglei (Emma) Fan [1,2]*

To develop active nanomaterials that can instantly respond to external stimuli with designed mechanical motions is an important step towards the realization of nanorobots. Herein, we present our finding of a versatile working mechanism that allows instantaneous change of alignment direction and speed of semiconductor nanowires in an external electric field with simple visible-light exposure. The light induced alignment switch can be cycled over hundreds of times and programmed to express words in Morse code. With theoretical analysis and simulation, the working principle can be attributed to the optically tuned real-part (in-phase) electrical polarization of a semiconductor nanowire in aqueous suspension. The manipulation principle is exploited to create a new type of microscale stepper motor that can readily switch between in-phase and out-phase modes, and agilely operate independent of neighboring motors with patterned light. This work could inspire the development of new types of micro/nanomachines with individual and reconfigurable maneuverability for many applications.

[1] Materials Science and Engineering Program and Texas Materials Institute, The University of Texas at Austin, Austin, TX 78712, USA. [2] Department of Mechanical Engineering, The University of Texas at Austin, Austin, TX 78712, USA. *email: dfan@austin.utexas.edu

The future micro/nanorobots require high degrees of freedom in motion control to perform complex tasks by individuals or by a swarm. However, it is a daunting task to apply the established technologies that manipulate macroscopic robots to realize similar functions in their micro/nanoscale counterparts. There are several grand challenges, ranging from design and fabrication of active nanocomponents, precision device assembling, to actuation with desired performances[1]. Furthermore, the physics that governs mechanical motions at micro/nanoscales is distinctive from that over the large scale. In addition to the level of required force to operate a micro/nanoobject being reduced by orders of magnitude, non-important forces in macroscopic regime, such as electrostatic interactions, become prominent in micro/nanoscale. Moreover, since most micro/nanomachines operate in aqueous suspensions, the related low Reynolds number physics[2], ionic effects[3], and interactions between suspension medium and energy sources emerge for consideration[4]. To address these challenges, in the past decades, rapid progress has been made in fundamental science[5,6], fabrication approaches[7,8], assembling strategies[9–16], and a variety of powering mechanisms of micro/nanomachines[17–22]. An array of potential applications has been demonstrated in biosensing, cargo and molecular delivery, single-cell bioresearch, and environmental remediation[23–27].

However, it remains a great challenge to control the motions of an individual nanomachine amidst many, to switch the operation modes facilely, and it is even more difficult to actuate several components of a nanomachine coordinately for purposed actions[28]. This high degree of versatility is essential for the future micro/nanorobots and requires investigation of innovative actuation mechanisms. Recently, research interest has been focused on utilizing two different propulsion mechanisms to generate stimulus-responsive motions of nanomachines that can alter among different types of motions[29]. For instance, catalytic and optochemical nanomotors can instantly align, stop, and even change propulsion directions in an external magnetic[30], or electric field (E-field)[31]. However, these demonstrations are largely based on combining two well-understood propulsion mechanisms to achieve switchable motion control.

It has been a rarely exploited concept to generate switchable operations of nanomotors among different modes via instantaneously tuning their physical properties, e.g., electric conductivity[32] with an external stimulus. Recently, our finding shows that visible light is able to change the imaginary part of the electric polarization of semiconductor nanowires, such as silicon, as reflected by the dramatic change of rotation orientation and speed in a high-frequency rotating E-field[32]. According to the Kramers–Kronig relation, the imaginary-part and real-part electric polarizations are correlated[33]. It suggests that if the imaginary part of the electric polarization alters with an external stimulus, the correlated real part could also respond to the same stimulus. However, the electrokinetic-related phenomena at low-frequency electric fields, e.g., < 1 kHz, including electroosmosis, electrolysis, and voltage screening, conceal the effect and thus hinder a strict implementation of Kramers–Kronig relation to predict the real-part polarization from the imaginary-part polarization (Supplementary Note 1). Therefore, it is fundamentally interesting to investigate and understand optical effects on the real-part polarization, and to demonstrate the associated unique applications.

Moreover, the optical tunable real-part (in-phase) polarization broadly impacts multiple distinct and more applicable types of electric manipulations, compared with that governed by the imaginary-part polarization[32]. Implications of changing the real part of the polarizability directly relate to at least three important manipulation phenomena— electro-alignment (studied in this work), dielectrophoresis (transport of particles with E-field gradient), and the chaining effect (particle–particle interactions). Studies in this work prepare a foundation for the improved controllability of these different types of manipulations, which can be exploited for an array of applications, such as light-tunable cargo transport and delivery, position-specific assembly, and collective operations of micro/nanorobots.

Electro-alignment refers to mechanical alignment of longitudinal particles in an AC E-field as a result of the interaction between the in-phase (real-part) electric polarization of the particles and the electric field. Here, we report our study of the light effect on the electro-alignment of silicon nanowires. When illuminated with visible light, a nanowire can be toggled between different alignment speeds and orientations. The switchable manipulation is instant, robust, and programmable, and can be utilized to communicate words in Morse code. Theoretical analysis and numerical simulation are carried out to reveal the underlying physical mechanism, which is attributed to the light-tunable in-phase (or real-part) electric polarization of semiconductor nanoparticles in suspension. The simulation results well agree with the experiments. With the obtained understanding, we propose and successfully realize the first light-switchable functional stepper micromotor with great improvement in performance compared with those in previous reports[34]. With designed light patterns projected by a digital light processing system (DLP), nanomotors demonstrate independent mechanical operations amidst neighbors in a same E-field.

## Results

**Optically tunable electro-alignment of silicon nanowires**. Silicon nanowires are fabricated via the well-known metal-assisted chemical etching (MACE) methods. The details are described in the Methods section. All the nanowires used in this study are ~10 μm in length and 500 nm in diameter, made from undoped silicon (Fig. 1a, b). Two types of microelectrodes made of Cr (5 nm)/Au (100 nm) films are patterned on glass substrates (Fig. 1c). The parallel microelectrodes are made of two rectangular metal pads separated by a uniform gap of 125 μm (Fig. 1d), which can generate a spatially uniform E-field in the center of the gap for measurement of the alignment rate of silicon nanowires from 5 kHz to 4 MHz and up to 15 $V_{pp}$ (Supplementary Note 2 and Supplementary Movie 1). The quadruple microelectrodes are made of four trapezoidal electrode pads, patterned orthogonally, creating a square central area of 500 μm in side length (Fig. 1e). They are utilized to actuate the light-controlled microscale stepper motor to be discussed later.

When placed in an AC E-field in the range of several kilohertz to megahertz, silicon nanowires can be instantly aligned in response to the E-field. This effect has been observed in longitudinal particles and is known as electro-alignment[35]. However, different from previous works, for Si nanowires, which normally align parallel to the applied E-field across a broad range of AC frequencies, we also observe perpendicular alignment to the E-field in a narrow range of frequencies in deionized water. Moreover, when illuminated with laser (532 nm), the alignment torques on silicon nanowires change remarkably, resulting in accelerated alignment speed and even switching between the parallel and perpendicular directions. We have successfully switched the alignment of one single Si nanowire between the two directions for hundreds of times, which, however, is not the limit as long as a wire maintains its position between the electrodes. This switchable manipulation has been observed for the first time. To understand this effect, we carry out a series of experiments, characterize the behaviors, and conduct theoretical analysis with numerical simulation.

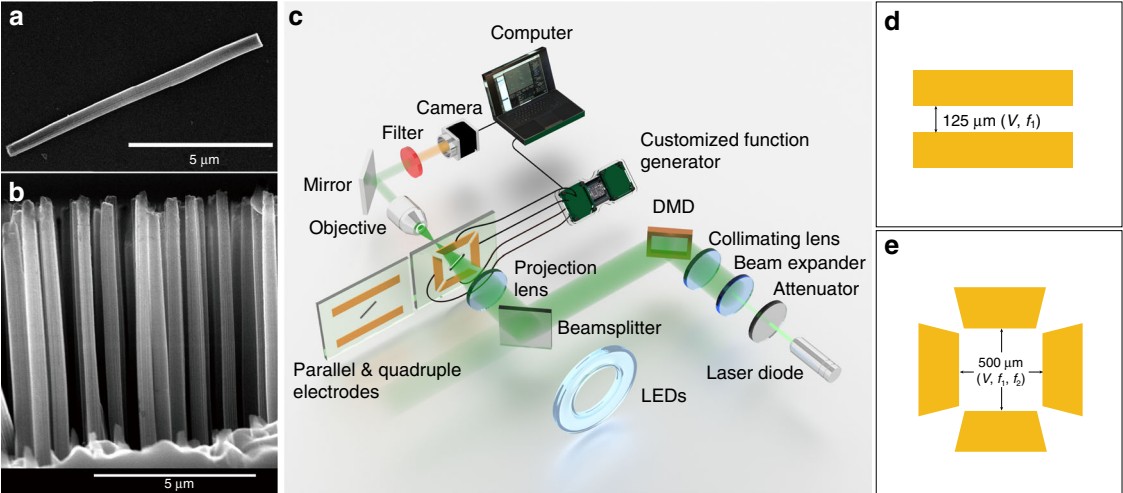

**Fig. 1 a, b** SEM images of the fabricated undoped silicon nanowires with 10-μm length and 500-nm diameter. **a** A single silicon nanowire. **b** Cross-section image of silicon nanowire arrays. **c** Schematic of experimental setup for switchable alignment of nanowires and rotation of micro stepper motors (relative position of each component is shown; the microelectrodes faces upward in the actual setup); (**d**) parallel microelectrodes for electro-alignment measurement; (**e**) and quadruple microelectrodes for actuation of the stepper motors.

It is known that when there is no external E-field, a nanowire suspended in water exhibits Brownian motions and is randomly oriented. For instance, at time $t_1$, a nanowire orients at $\theta_1$ ($\theta$ is the angle between the long axis of the nanowire and the E-field). As soon as an E-field is applied via the parallel microelectrodes, the nanowire rotates to align in response to the E-field. At time $t_2$, the angle turns to $\theta_2$. The alignment rate ($A$) can be determined as shown in Supplementary Note 3:

$$A = -\frac{1}{t_2 - t_1} \int_{\theta_1}^{\theta_2} \frac{\mathrm{d}\theta}{\sin\theta\cos\theta} \qquad (1)$$

Most frequently, the nanowire aligns parallel to the E-field with $\theta \approx 0$, which is termed as parallel alignment with a positive alignment rate. Interestingly, the nanowire also aligns perpendicular to the E-field with $\theta = \pi/2$ under certain conditions, which is termed as transverse alignment with a negative alignment rate.

By sweeping the AC frequency at an E-field of 1200 V cm$^{-1}$ (15 Vpp), and toggling the laser on and off, we obtain the alignment behaviors of Si nanowires at different frequencies with (red curve) and without laser (blue curve) in Fig. 2a. Here, a minimum LED illumination (white light ~500 lx) is used as background lighting to record motions of nanowires when there is no laser. A 532-nm laser at an intensity of 32 mW cm$^{-2}$ is used to illuminate nanowires together with the dim LED to obtain the laser-induced alignment behaviors in the red curve.

With the minimum background LED light and no laser exposure, the frequency-dependent alignment of a nanowire can be categorized into three regimes: (1) in the low-frequency regime from 5 kHz to 500 kHz, nanowires align parallel to the applied E-field with a maximum alignment rate of 150 rad s$^{-1}$ at 5 kHz. The rate gradually decreases with AC frequency with a steep drop at ~100 kHz; (2) in between 500 kHz and 1 MHz, the nanowires are aligned perpendicular to the E-field; (3) when the AC frequency further increases to above 1 MHz, the nanowires return to the parallel alignment at a low rate below 10 rad s$^{-1}$. In contrast, when the 532-nm laser is illuminated on the silicon nanowires, dramatic changes of the alignment behaviors, including acceleration and even switching of direction are observed. For instance, at 5 kHz, light illumination greatly increases the alignment rate by 2.4-folds from 150 rad s$^{-1}$ to 360 rad s$^{-1}$. The

maximum alignment rate increases to 487 rad s$^{-1}$ occurring at 25 kHz, which is almost four times that of the same frequency without laser. As the frequency further increases, the alignment rate decreases rapidly at ~250 kHz and verges to zero at 4 MHz. Overall, laser exposure increases the positive alignment rate at all frequencies, blue-shifts the frequency where the highest alignment rate occurs, and changes the transverse alignment to parallel alignment at the tested frequencies.

To further understand the observed light-controlled switching behaviors of Si nanowires, we monotonically vary the laser intensity from 8 mW cm$^{-2}$ to 127 mW cm$^{-2}$ and measure the alignment rate as shown in Fig. 2b. For instance, at 5 kHz, the alignment rate increases with laser intensity up to 32 mW cm$^{-2}$, then maintains an approximate plateau to 64 mW cm$^{-2}$, before decreasing at 127 mW cm$^{-2}$. The observed decrease of alignment rate under stronger laser stimulus will be further studied and discussed in the following with simulations. In general, the maximum alignment rate among all tests increases with laser intensity and the corresponding peak frequency exhibits a monotonic shift from 5 kHz at 0–16 mW cm$^{-2}$ to 100 kHz at 127 mW cm$^{-2}$. The amplitude of the negative alignment rate decreases with laser power up to 16 mW cm$^{-2}$, above which the transverse alignment switches to parallel alignment. In order to provide more insights, we also calculated the corresponding anisotropy of polarizability of nanowires as shown in Fig. 2a, b (axes on the right, details in Supplementary Notes 4, 5).

**Light programmable toggle of nanowires between two alignment orientations to express words in Morse code.** Finally, we use this effect as a light-controlled parallel-to-transverse alignment switch. We expose a nanowire in the E-field of 750 kHz and 1200 V cm$^{-1}$, and to the 111 mW cm$^{-2}$ 532-nm laser, which is periodically occluded via a motorized shutter. The nanowire can be switched instantly between parallel and transverse alignment to the E-field, when the laser is on and off, respectively. The switch periodicity follows the on/off state of the laser as shown in Fig. 2c (see Supplementary Movie 2), demonstrating the robustness of the light-controlled switch of semiconductor nanowires in an AC E-field. The high controllability and reproducibility of the light-controlled alignment are further exploited by toggling a nanowire to express words, i.e., "HELLO WORLD", in Morse code

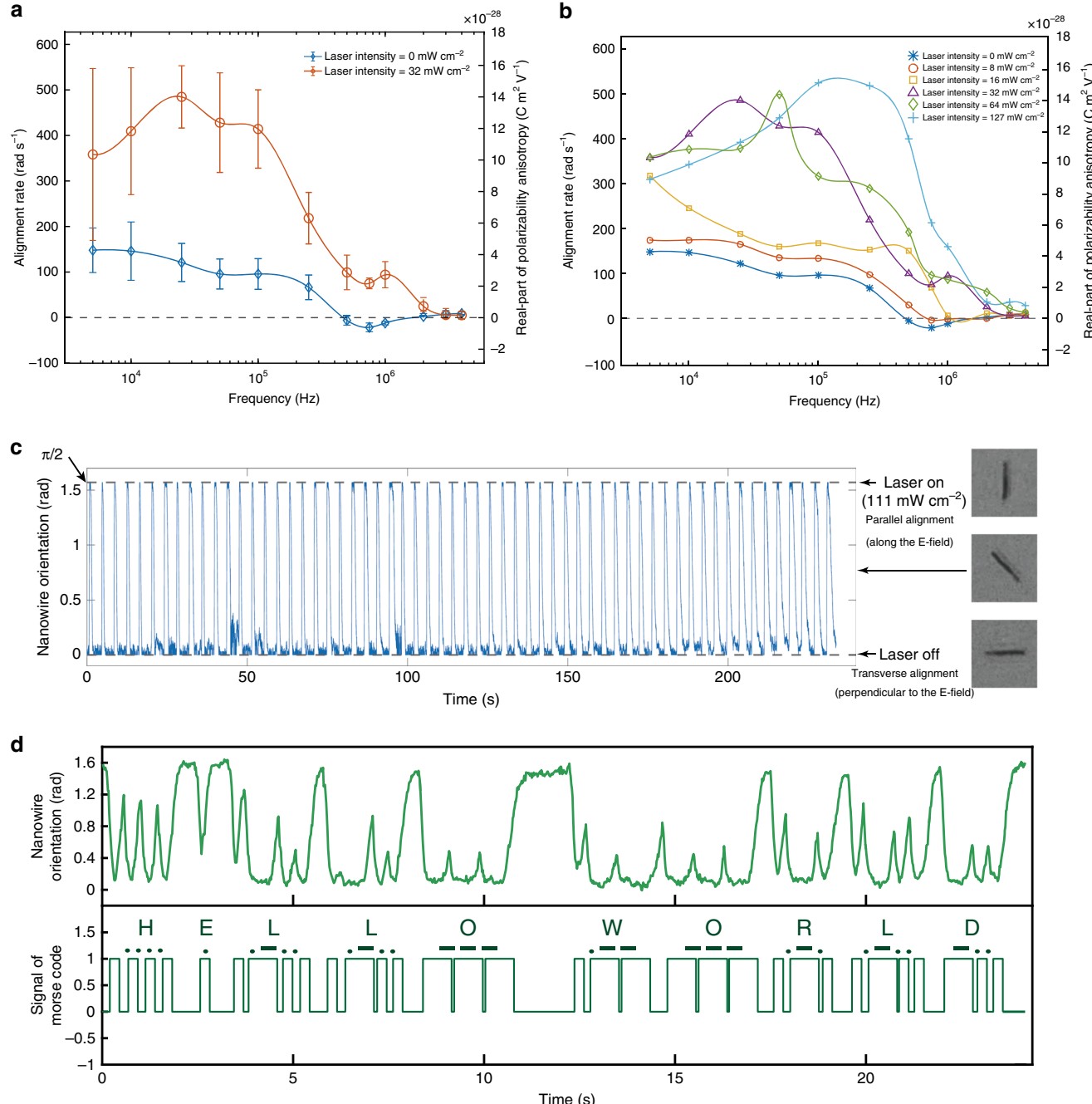

**Fig. 2** Light-responsive alignment of Si nanowires. **a** Alignment rate versus AC frequency of silicon nanowires (undoped; length, 10 μm; diameter, 500 nm) in the dim condition and under 532-nm laser illumination (error bars are defined as s.d., source data are provided in a Source Data file). **b** Intensity effect of laser on alignment rate versus AC frequency. The corresponding anisotropy of polarizabilities are shown on the right axes, and the eye-guiding lines in (**a**) and (**b**) show trends. **c** Cyclic switch between parallel and transverse alignment of a single nanowire with on/off of a 532 nm laser. The E-field direction is defined as π/2, and its transverse direction is defined as 0. **d** Toggling of a single nanowire in response to a light signal encoded by Morse code (top), and the corresponding interpreted words "HELLO WORLD" from the mechanical motions of the nanowire (bottom).

(International Telecommunication Union standard). Here, the duration of the alignment towards and at 90° when the laser is off is used to transmit Morse code, i.e. a short signal of dot (·) takes a third of the time of a long signal of dash (-). When the laser is on, the corresponding duration of alignment motion toward and at 0° forms the separation space between signals. Nanowires switch back and forth with controlled durations in response to the programmed light signals according to the Morse code (Fig. 2d, Supplementary Movie 3, and Supplementary Data 1). The toggled motions well correlate with the light signals, and can be

interpreted as "HELLO WORLD" (Supplementary Data 2). This demonstration is one of the first that utilizes mechanical motions of a nanowire to transmit meaningful words, which could be a potential communication method of the future nanorobots.

## Modeling of the optically tunable in-phase electric polarization of silicon nanowires. To understand the above unique light-controlled alignment, we carry out theoretical analysis followed by numerical simulation and modeling. In the presence of a uniform AC E-field, an electric dipole moment (**p**) is induced in a silicon

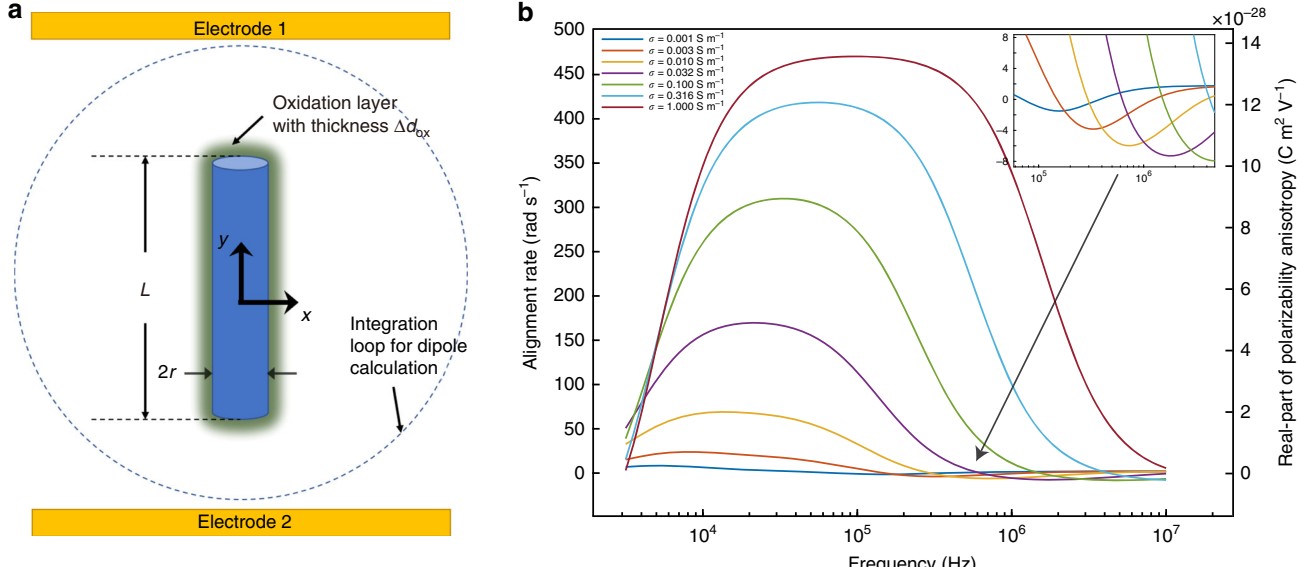

**Fig. 3** Modeling of nanowire polarization and numerical calculation of alignment rate versus AC frequency of a silicon nanowire of $L = 10\,\mu m$ and $r = 250$ nm. **a** Model of a silicon nanowire as a cylinder with length $L$ and radius $r$. An Oxidation shell with thickness $\Delta d_{ox} = 1$ nm is considered, shown as the dark shade surrounding the nanowire. **b** Numerical calculation of alignment rate of nanowires of different electrical conductivities versus AC frequency. Inset: zoom-in image at frequencies with negative alignment rates.

nanowire. When the dipole moment is directed at an arbitrary angle to the E-field, an electric torque, given by $\boldsymbol{\tau} = \mathbf{p} \times \mathbf{E}$, exerts on the nanowire. For simplicity, we consider only a 2D case, where the motions and the involved forces are both in-plane with the microelectrodes. We can express the E-field along the x-axis with $\mathbf{E} = \mathrm{Re}[E_0 \hat{\mathbf{x}} \exp(i\omega t)] = \mathrm{Re}[\tilde{\mathbf{E}} \exp(i\omega t)]$, where $\tilde{\mathbf{E}} = E_0 \hat{\mathbf{x}}$ is the phasor of E-field, $E_0$, $\omega$, and $t$ are the amplitude, angular frequency of the E-field, and time, respectively. The dipole moment ($\mathbf{p}$) of the nanowire can be decomposed into two components that are parallel and perpendicular to the long direction of the nanowire, given by $\mathbf{p}_\parallel = \alpha_\parallel \tilde{\mathbf{E}}_\parallel$ and $\mathbf{p}_\perp = \alpha_\perp \tilde{\mathbf{E}}_\perp$, respectively. Here $\mathbf{p}_i, \alpha_i, \tilde{\mathbf{E}}_i (i = \parallel \text{ or} \perp)$ are the phasor of dipole moment, complex polarizability, and the phasor of the E-field along the respective directions. Since the frequency of the AC E-field employed in our experiments is much higher than that of the rotational motion during nanowire alignment, the time-averaged torque ($\tau$) exerted on a nanowire results in the observed alignment, which can be expressed as:

$$\boldsymbol{\tau}_e = \frac{1}{2}\mathrm{Re}[\mathbf{p} \times \tilde{\mathbf{E}}^*] = -\frac{1}{2}E_0^2 \mathrm{Re}\left(\alpha_\parallel - \alpha_\perp\right)\sin\theta\cos\theta\hat{\mathbf{z}}, \quad (2)$$

where $\theta$ is the angle between the long axis of the nanowire and the E-field, and $\tilde{\mathbf{E}}^*$ denotes the complex conjugation of the E-field phasor.

Owing to the small size, a nanowire is in an extremely low Reynolds number regime. As a result, the drag torque is proportional to the rotation speed. The driving electric torque on the nanowire is balanced by the liquid drag torque essentially instantly, given by $\tau_e = -\tau_{\mathrm{drag}} = \gamma\dot{\theta}$, where $\gamma$ is the rotational drag coefficient of a nanowire in deionized water. As a result, the alignment speed of a nanowire depends on the instantaneous angle ($\theta$) between the long axis of the nanowire and the E-field, and is given by:

$$\dot{\theta} = -A\sin\theta\cos\theta, \quad (3)$$

where $A = \frac{E_0^2}{2\gamma}\mathrm{Re}(\alpha_\parallel - \alpha_\perp)$ is the alignment rate. Note that Eq. (3) indicates that the nanowire rotation speed is zero at angles $\theta = 0$

or $\pi/2$ and half of $A$ when $\theta = \pi/4$. The alignment rate ($A$) can be experimentally determined by Eq. (1). Furthermore, since $A$ is linearly proportional to $\mathrm{Re}(\alpha_\parallel - \alpha_\perp)$, one can determine the real part of the anisotropy of polarizability of a nanowire at a given electric field strength ($E_0$) and viscous drag coefficient ($\gamma$). A positive alignment rate ($A > 0$) indicates $\mathrm{Re}(\alpha_\parallel - \alpha_\perp) > 0$, where the real part of electric polarization along the parallel direction of a nanowire is greater than that along the transverse direction. Vice versa, a negative alignment rate ($A < 0$) corresponds to $\mathrm{Re}\left(\alpha_\parallel - \alpha_\perp\right) < 0$.

To obtain a more qualitative understanding of the optically controlled alignment phenomena, we carry out modeling with both analytical and numerical approaches. Related to this work, modeling of the out-of-phase (imaginary part) polarization of semiconductor nanoparticles, has been reported by using either a numerical method[36] or an analytical approach[32]. Different from previous works, here, we focus on the in-phase electric polarization (real part) and leverage the accuracy of numerical simulation and simplicity of analytical approximation to investigate the system.

We consider two major frequency-dependent effects in the model, Maxwell-Wagner interfacial polarization[35] and the electrical double-layer (EDL) charging[36]. Maxwell-Wagner polarization originates from the difference between electrical properties of a particle and the suspension medium, which results in a net electric dipole moment in the particle[35]. With the well-established theory of Maxwell-Wagner polarization, the frequency-dependent dipole moment of the nanowire can be calculated analytically. However, in the analytic models of a nanowire, the shape is approximated as a prolate ellipsoid. With numerical simulation based on finite element analysis, we can take the accurate wire shape as a cylinder in the model as shown in Fig. 3a, implemented with COMSOL. Furthermore, it is known that as-fabricated Si nanowires inevitably has a native SiO$_2$ layer. Here, we can include an oxidation layer of 1-nm on the silicon nanowire in the model to reflect the nature of the system. The Maxwell-Wagner polarization contributed dipole moment is calculated by a integration method[37] (Supplementary Note 4).

Next, we model the electrical double layer around the electrically polarized nanowire. An equivalent resistor-capacitor (RC) model is used to predict the frequency-dependent behavior[32]. Since the EDL is formed by ions in aqueous suspension with finite mobilities, there is a phase lag ($\delta$) between the Maxwell-Wagner polarization and EDL charging. In a low-frequency E-field, ions can move rapidly enough to make the phase lag $\delta$ small, resulting in an opposite dipole moment to the direction of Maxwell-Wagner polarization. At higher frequencies, the phase lag gradually increases, and eventually, the effect of the EDL vanishes since the ionic diffusion speed cannot catch up the oscillation frequency of Maxwell-Wagner polarization. Therefore, we can calculate the imaginary and real parts of the EDL charging dipole as follows:[32]

$$\text{Im}(P_{\text{EDL}}) = -\frac{[\text{Re}(P_{\text{MW}})\sin\delta + \text{Im}(P_{\text{MW}})\cos\delta]}{\sqrt{\omega^2\tau_{\text{RC}}^2 + 1}} \text{ and}$$
$$\text{Re}(P_{\text{EDL}}) = -\frac{[\text{Re}(P_{\text{MW}})\cos\delta - \text{Im}(P_{\text{MW}})\sin\delta]}{\sqrt{\omega^2\tau_{\text{RC}}^2 + 1}} \quad , \quad (4)$$

respectively, where $P_{\text{MW}}$, $\delta$, $\omega$, and $\tau_{\text{RC}}$ are the dipole moment of Maxwell-Wagner polarization, phase lag between polarization and EDL charging that follows $\tan\delta = -\omega\tau_{\text{RC}}$, angular frequency of the E-field, and time constant of the RC model. The total electric dipole moment is the summation of both Maxwell-Wagner and EDL polarizations, given as $P_{\text{total}} = P_{\text{MW}} + P_{\text{EDL}}$.

We simulate the alignment rate and the anisotropy of electrical polarizability to compare with the experimental results as shown in Fig. 3b. Overall, the simulation agrees with experiments. The respective contributions of polarization from Maxwell-Wagner relaxation and electrical double layer are provided in Supplementary Note 5 and Supplementary Figure 1. In simulation, the maximum alignment rate and peak frequency increase with the electrical conductivity of silicon nanowires, corresponding to the increase of alignment rate with laser intensity in experiments. Importantly, this model also predicts the transverse alignment at ~750 kHz when the conductivity of silicon is $1 \times 10^{-2}$ S m$^{-1}$, which agrees with that observed experimentally. As aforementioned, we observed decrease in alignment rate when light intensity increases at low frequencies, i.e. 5 kHz. This is also found in the simulation: the alignment rate decreases with increase of electrical conductivity in Fig. 3b (< 5 kHz). Based on this model, we can understand the observed phenomenon with an intuitive physics picture involving the capacitive effect of oxide at low frequencies (Supplementary Note 6 and Supplementary Fig. 2).

However, there is also a difference between simulation and experiments. With the increase of the electrical conductivity resulted from light illumination, the simulation shows that transverse alignment exhibits a monotonical increase of alignment rate and a blue-shifted peak frequency. This differs from our experimental results, where the transverse alignment vanishes when the laser intensity is higher than 16 mW cm$^{-2}$. This could be due to mechanisms that have not been understood, which require further investigation.

**Light programmable synchronous stepper micromotor.** With the understanding of the optically tunable electro-alignment of semiconductor nanowires, we propose an innovative light-controlled synchronous stepper micromotor. It mimics the macroscopic synchronous stepper motors, which rotate in-phase with an E-field and turn to specific angles on demand. The operation phase of our miniaturized motors can be switched by light. To realize such a micromotor, we generate a high-frequency AC E-field ($f_1 = \frac{\omega_1}{2\pi}$) that efficiently aligns a nanowire and rotates this high-frequency AC field at a much lower frequency

($f_2 = \frac{\omega_2}{2\pi} \ll f_1$) clockwise, so that the nanowire follows the low-frequency rotating AC field with a continuous synchronous rotation. Such an E-field can be given by:

$$\mathbf{E} = E_0 \cos(\omega_1 t) \cdot [\cos(\omega_2 t)\hat{\mathbf{x}} + \sin(\omega_2 t)\hat{\mathbf{y}}]. \quad (5)$$

We note this driving mechanism of the motor differs from most previous works, where the rotation is a result of the interaction between a high-frequency rotating E-field and out-of-phase electric polarization of a nanowire[32,34]. Here, an in-phase electro-alignment torque is created and continuously compel the rotation of micromotors[38] with the advantages of precision speed control and angular positioning, synchronism among different motors, and programmable light switching. Specifically, the created stepper micromotor in this report exhibits at least three distinct advantages compared with those in previous works: (1) the time-dependent angular position of a motor can be accurately controlled by the applied E-field, without the use of imaging feedback. (2) Under proper lighting conditions, all the motors, regardless of their differences in geometry or electrical property, can be aligned in the same direction synchronously, which makes this mechanism a potential approach to control rotation of arrays of motors at exact same speed under the same field. (3) The alignment field in this work can provide an effective torque to fix the motor's angle when it is stopped, countering the Brownian motion.

The driving E-field is created by applying four AC voltages as shown in the following equations sequentially on the quadruple microelectrodes in Fig. 1e:

$$\begin{aligned} V_{1,3} &= \pm V_0\sin(\omega_1 t)\cos(\omega_2 t) \\ V_{2,4} &= \pm V_0\sin(\omega_1 t)\sin(\omega_2 t) \end{aligned} \quad (6)$$

The voltages can be generated via a customized computer-controlled four-channel function generator (Supplementary Note 2). In such E-field, we observe that a micromotor always operates in one of two modes, the "in-phase" or "out-phase" modes, depending on whether the alignment torque is enough to overcome the drag. The motor runs in-phase when it rotates at the same rotating speed of $\omega_2$ ("synchronous speed") with the rotating E-field, where the alignment torque satisfies: $\tau_e = -\tau_{\text{drag}} = \gamma\omega_2$. As previously discussed, the alignment torque is a function of the angle between the nanowire and the E-field, given by: $\tau_e = -\gamma A \sin\theta\cos\theta$, so the maximum alignment torque $\tau_e$ available to drive the motor is $\frac{\gamma A}{2}$, which occurs when the angle between the nanowire long axis and the E-field is $\pi/4$. Therefore, if $\frac{\gamma A}{2} \geq \gamma\omega_2$, the E-field can supply a sufficient torque to power the motor's in-phase rotation as shown in Fig. 4a. In particular, the nanowire rotates at an approximately constant angle $\theta$ behind the E-field, where $\theta$ is termed as the "phase lag" (Fig. 4a inset). Otherwise, the alignment torque is insufficient and the nanowire, though rotates, periodically falls behind the E-field resulting in oscillations as shown in Fig. 4b. When the motor runs out-phase, it exhibits a net speed, in the same direction but lower than the rotating E-field. We find attractive features in these out-phase oscillations. More analysis is provided in Supplementary Note 7 for readers' interest.

Laser illumination controls the motion of such a micromotor and allows fine control beyond basic electric manipulation. To investigate the optical effect on the rotation of the micromotor, we scan the peak-to-peak voltage from 4 Vpp to 28 Vpp for an E-field rotating at a constant speed of $f_2 = 1.49$ Hz. Without laser exposure, the micromotor always rotates out of phase. While its average speed increases with the voltage, it is always lower than $f_2$. When excited by a 318 mW cm$^{-2}$ 532-nm laser and the voltage is above or equal to 14 Vpp, the micromotor switches from the

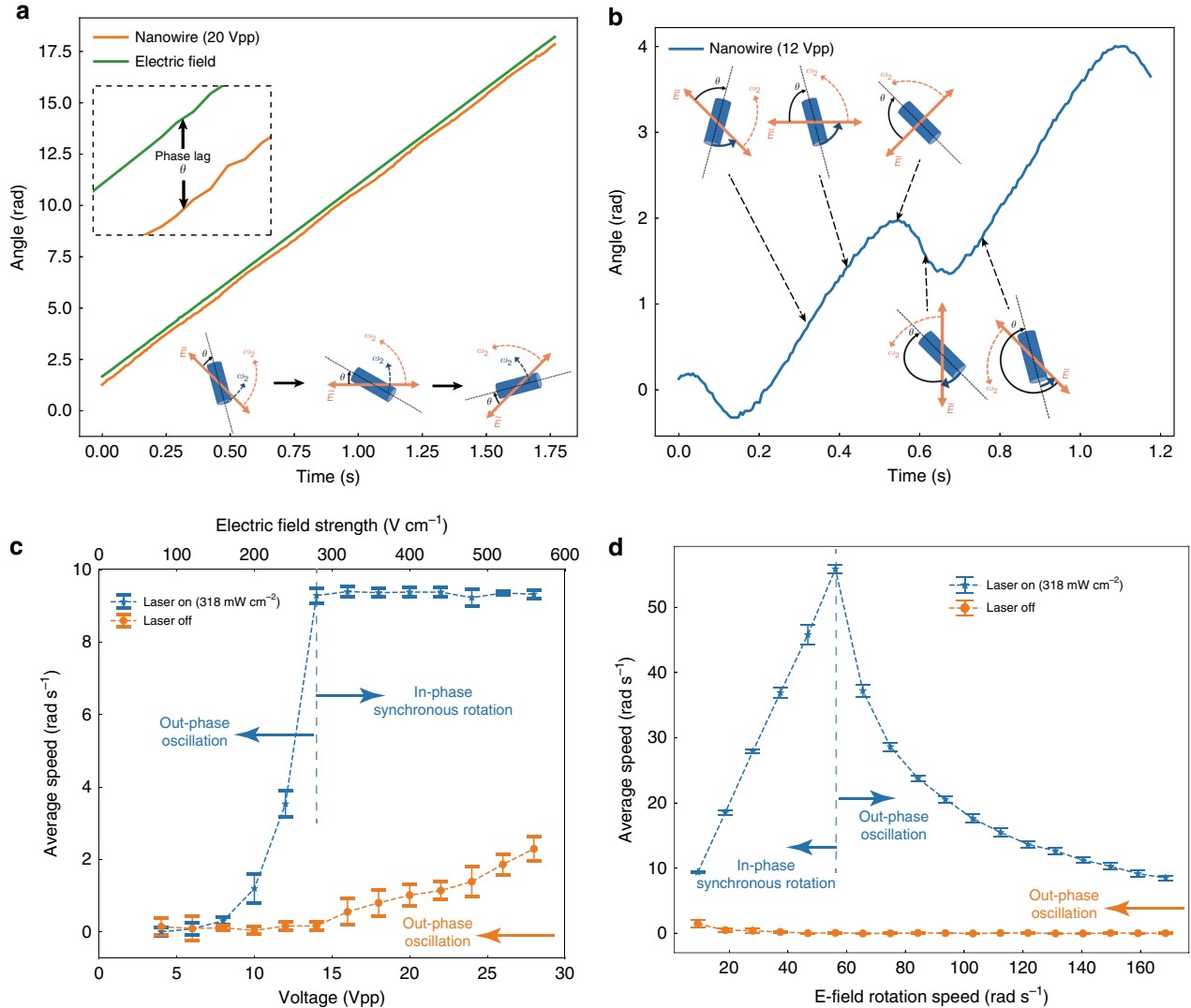

**Fig. 4** Light-switchable rotation of a Si nanowire stepper motor. **a**, **b** Rotation angle versus time of a nanowire at different voltages (or electric field strengths) exposed to 318 mW cm$^{-2}$ 532-nm laser. **a** The E-field rotates at 1.49 Hz (in green). The stepper motor rotates synchronously with the E-field with a phase lag $\theta$, at 20 Vpp (400V cm$^{-1}$) (in orange). Inset: the constant phase lag between the driving E-field and the rotating nanowire. **b** It switches to the out-phase oscillation mode at 12 Vpp (240 V cm$^{-1}$) (more details in Supplementary Note 7 and Supplementary Figs. 3, 4). Schematics show the relative positions of a rotating nanowire with the E-field at different time. **c** Average rotation speed of a micromotor versus driving voltage/electric field strength with (in blue) and without laser illumination (in orange). The driving E-field is 1.49 Hz, same as that the in-phase rotation of the nanowire (see Supplementary Movies 4, 5 for the curve in blue). **d** Average rotation speed of a micromotor versus rotation speed of the driving E-field at 30 Vpp (600 V cm$^{-1}$) with (in blue) and without laser illumination (in orange) (see Supplementary Movies 6, 7 for the curve in blue).

mode of out-phase oscillation to in-phase synchronous rotation at the same speed of the E-field ($f_2$) (Fig. 4c).

We further scan the rotation speed of the E-field ($f_2$) from 1.49 Hz to 26.82 Hz at a constant voltage of 30 Vpp. Without laser illumination, the micromotor always operates out-of-phase oscillation with nearly zero average speed. With a 318 mW cm$^{-2}$ 532-nm laser stimulation, the motor switches to in-phase synchronous rotation up to 8.94 Hz before losing the synchronism and turning into out-of-phase oscillation (Fig. 4d). The characterizations of the performances of the stepper motors in the mode of synchronous operation, such as the pull-out torque and power, are included in the Supplementary Note 8 and Supplementary Fig. 5. The two operation modes can be switched back and forth with simple control of laser exposure (see Supplementary Movies 8, 9). Furthermore, by adjusting the laser intensity, it is possible to finely tune the pull-out torque and power of the motor. The performance of the motors is well

reproducible (Supplementary Note 9 and Supplementary Fig. 6) and can be observed for more than a week with gradually increased threshold voltage (Supplementary Note 10 and Supplementary Fig. 7).

Leveraging the unique light responsiveness, we successfully operate micromotors independently and versatilely in the same E-field by projecting localized light spots via a digital light processing (DLP) system (Supplementary Note 2). The DLP system is capable of projecting microscale light patterns onto the sample surface with a maximum resolution of 5 μm per pixel. To control two micromotors individually, we programmed two circular laser spots of 70 μm in diameter to cover each micromotor without overlap (Supplementary Movie 10). A rotating E-field with $f_1 = 100$ kHz, $f_2 = 1.49$ Hz at 13 Vpp is applied as the driving source. When both laser spots are on, both micromotors rotate synchronously at the same speed, as shown in Fig. 5a, where the first segments of orange and blue curves

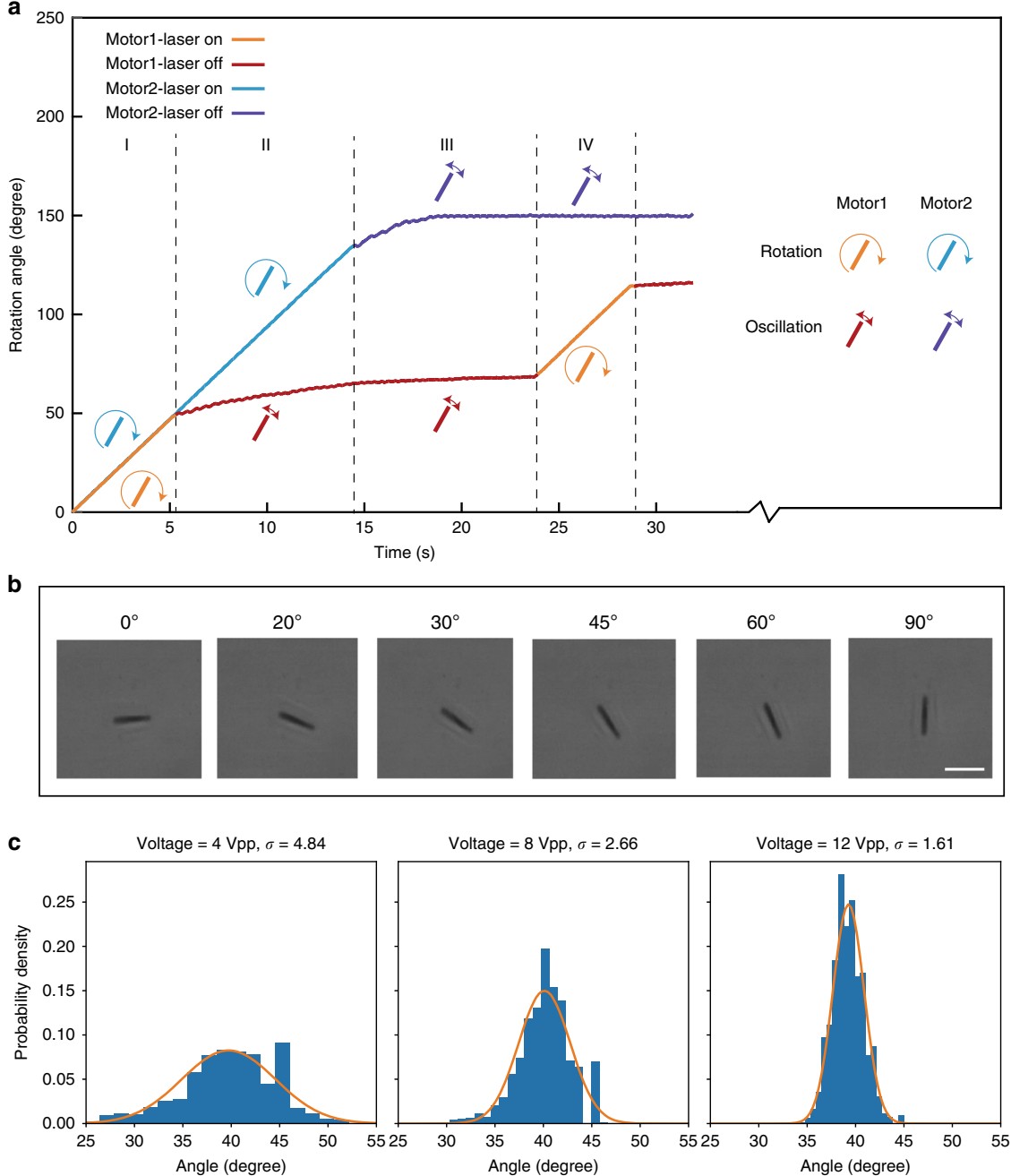

**Fig. 5 a** Independent control of two stepper micromotors in the same E-field. The micromotors are both driven by a rotating E-field with $f_1 = 100$ kHz and $f_2 = 1.49$ Hz at 13 Vpp. Digital light projecting (DLP) system projects light on each micromotor independently at an intensity of ~50 mW cm$^{-2}$. **b** Rotation of a micromotor to designated angular positions like a stepper motor by stopping the rotation of the AC E-field at specific angle ($f_1 = 100$ kHz, V = 10 Vpp, laser intensity = 127 mW cm$^{-2}$), scale bar 10 μm. **c** Angle fluctuation around a designated angle at different voltages (frequency = 100 kHz). Probability density is fitted by normal distribution.

overlap, indicating the synchronous rotation speed of motor 1 and 2. Then the laser spot on motor 1 is turned off. The driving torque on motor 1 is no longer sufficient to maintain the synchronism with the E-field, and the motor 1 steps into out-phase oscillation with a lower average speed (red curve). Next, the laser spot on motor 2 is also turned off; instantly, the motor 2 is switched into out-phase oscillation (purple curve). When both laser spot 1 and 2 are turned on again, both motors restore to synchronous rotation. To the best of our knowledge, this is the first demonstration of microscale stepper motors that can be operated on individuals independently in the same environment.

The success is based on modulating the electrical properties of each nanoparticle device instantaneously with external stimuli.

Finally, such micromotors can programmably step to designated angular positions as shown in Fig. 5b and Supplementary Movie 11. Here, the E-field (100 kHz, 10 Vpp, 200 V cm$^{-1}$) is directed to various angles by continuing its high-frequency component ($f_1$), but pausing the low-frequency component ($f_2$) at a designated angle. As shown in Eq. (2), our step motor can provide a restoring force of $\tau_{res} = A(\omega, E_0) \sin\theta \cos\theta$, in which the alignment rate A is a function of AC frequency and the electric field strength. It is obvious that the alignment accuracy

increases with the electric field strength and polarizability at an AC frequency. Here, we investigate the angle fluctuation of a stepper motor at different voltages as shown in Fig. 5c. Under 100 kHz AC E-field, when the voltage increases from 4 Vpp (80 V cm$^{-1}$) to 12 Vpp (240 V cm$^{-1}$), the normal distribution of angle of the stepper motor becomes narrower with the standard deviation $\sigma$ reducing from 4.84° to 1.61°.

## Discussion

In summary, we report a versatile working mechanism that can be exploited for developing a new type of efficient visible-light-responsive micro/nanomachines without requirements of chemical fuels, UV light source, or special geometry of nanoparticle building blocks. The electrical property of a semiconductor micromotor can be controlled solely by intensity of visible light, and is exhibited as modulable mechanical alignment in an external E-field (Supplementary Note 11). Based on our understanding, such an effect can be generalized to any types of materials as long as their electrical conductivities can be changed prominently when stimulated by light or any other stimuli. Various types of inorganic and organic photoconductive materials may all exhibit such an effect. Here, we demonstrated with the most common type of semiconductor, Si. The specific performance depends on the band structure, carrier lifetime, light wavelengths and intensity, dimension of the particle, and electrical properties of the suspension medium. In terms of other types of stimuli, we consider that materials that change electrical conductivities in magnetic fields, i.e., those with high magnetoresistance[39], could possibly exhibit an analogous effect.

The facile switching mechanism is a new addition to the tool box of nanomanipulation techniques, and could offer opportunities that are worthy of exploration. For instance, as shown in the demonstration, encoded light signals can be transmitted into alignment oscillations of a single device, which communicates meaningful words in Morse code. The same approach could be potentially explored to develop unique opto-mechanical conversions for complex coupled micro/nanomachines. Moreover, with patterned light from DLP, we successfully demonstrated individually controlled rotary stepper motors in the same field, each can toggle between the oscillation and full-turn synchronous rotation modes, independently. With tunable intensity of light, it is feasible to control each in the pull-out torque and power as shown in the Supplementary Note 8. Ultimately, it could be attainable to control arrays of stepper micro/nanomotors that exhibit individually controlled performances by dynamic light patterns. The working mechanism could also be exploited for the future micro/nanorobots independently operating for collaborative task forces, and even to control different components within a micro/nanorobots for programmable operations.

In the aspect of fundamental research, the light effect on the real part of the electrical polarization anisotropy is studied for the first time. We investigate the light stimulated electroalignment behavior of Si nanowires with experimentations and simulations, and demonstrated applications. This work also prepares the foundation for other types of related phenomena, i.e., light-tunable transport and chaining of particles, and adds the knowledge of the optical effect on the real part of electrical polarizability anisotropy of Si nanowires in aqueous solution.

Every coin has two sides. The presented switchable manipulation is efficient and versatile. However, it is more feasible for on-chip devices than in-vivo applications at the present stage. Also, a few challenges should be addressed in the future, including the oxidation problem that many semiconductors have in aqueous solution, and the compatibility with complex media[40,41]. More sophisticated device design should be considered to replace the simple wires for useful applications. The full potential of this technique will also be helped by advanced optical projection devices with high resolution and large projection area, as well as sophisticated programmability to equip automation[42] and intelligence to the manipulation. Overall, this work suggests new opportunities for the emerging nanorobotics in both the fundamental and application aspects.

## Methods

**Fabrication of silicon nanowires**. We fabricate silicon nanowires via the well-known metal-assisted chemical etching (MACE) methods as previously discussed[32] with a slight modification. In brief, a dispersed monolayer of 500 -nm diameter polystyrene nanospheres is assembled on a cleaned undoped silicon wafer. A catalytic metal thin film of 25 nm Ag and 5 nm Au is deposited by electron beam evaporation on the top such that the polystyrene nanospheres act as a mask for the following fabrication. A scotch tape is used to remove the nanospheres, leaving a metal film with circular nanoholes on the wafer. Immersing the sample into the etchant composed of 4.7 M hydrofluoric acid and 0.3 M hydrogen peroxide dissolves the silicon underneath the metal film to leave arrays of nanowires. Finally, we use silver and gold etchants to remove the catalytic metal layer, followed by sonication in DI water to break the nanowires off the substrate. All the nanowires used in our experiments are ~10 μm in length, 500 nm in diameter, and composed of undoped silicon. The nanowires are stored in DI water and used in a few days after preparation. A surface oxide layer develops in a few days.

**Fabrication of microelectrodes**. We also make two types of electrodes, "parallel" and "quadrupole", for measurement of nanowire-to-E-field alignment rate and nanowire micromotor control, respectively. The electrodes are made of 5 -nm Cr and 100 -nm Au thin film deposited via electron beam evaporation in one of two patterns (via standard photolithography), shown in Fig. 1d, e, onto a glass microscope slide. The parallel electrodes are two large rectangular pads separated by a long 125 μm gap, while the quadrupole electrodes are four rectangular pads surrounding a square area 500 μm on a side (simulation of E-field distribution is included as Supplementary Note 12 and Supplementary Fig. 8). The parallel electrodes are used for measuring nanowire-to-E-field alignment rates because the electrode configuration creates a spatially uniform E-field in the center, thereby minimizes dielectrophoretic effects, while the quadrupole electrodes are used to change the angle of the E-field to create nanowire micromotors. The parallel and quadrupole electrodes are attached via silver epoxy to an Agilent 33250 A function generator (capable of 5 kHz to 4 MHz signals up to 15 Vpp) and a customized computer-controlled four-output function generator (capable of arbitrary waveforms up to 2 MHz and 30 Vpp), respectively. All reported voltages assume voltage drops in the nanowire suspension between electrodes. The electric field strength is the division of the voltage by the electrode separation distance.

**Experimental setup**. In each experiment, nanowires in suspension of ~20 μL is placed on top of the electrodes (either parallel or quadrupole) inside a PDMS microwell covered by a class slide. Nanowires gradually deposition to the bottom of the setup and stay in-plane with the microelectrodes. A 532 nm diode-pumped solid-state laser (Thorlabs) is used for light excitation, which can be toggled off and on at different intensities. In order to reduce the influence of background light, the sample is illuminated with an always-on custom LED white light source of ~500 lx. A Basler acA1300-200 μm camera captures images of the electrodes and nanowires (with laser filtered out) at up to 1280 × 1024 pixels and up to 1000 frames per second (FPS). A computer program can track nanowire positions for data analysis via standard computer vision algorithms (Supplementary Movie 1), and vary the voltage amplitude from the four-output function generator in real time to produce a rotating E-field (Supplementary Note 2).

## Data availability

All relevant data are available from the authors upon request, relevant Matlab codes are available as supplementary materials.

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

## Acknowledgements

We are thankful for Prof. Peer Fischer (Max Planck Institute for Intelligent Systems and University of Stuttgart, Germany) and Prof. Ambarish Ghosh (Indian Institute of Science, India) for the helpful discussions. We would also like to thank S. Miao and K. Seong for the technical support. We are grateful for the support of NSF via the CAREER Award (grant no. CMMI 1150767 and intern supplement) and research grants (1710922 and 1930649 in part) and the support of the Welch Foundation (grant no. F-1734).

## Author contributions

D.F. and Z.L. conceived the research project. D.F. supervised the project. D.T. designed and implemented the customized function generator and the computer vision tracking system. Z.L designed and setup the experimental apparatus, implemented the modeling. D.T. and Z.L. conducted data analysis. All authors analyzed and discussed the results and co-wrote the paper.

## Competing interests

The authors declare no competing interests.
