## [Peer Review File · Nature Communications]

Reviewer #1 (Remarks to the Author):

The paper "Light Programmable Micro/Nanomotors with Optically Tunable In-Phase Electric Polarization" by Liang, Teal and Fan describes the effect of light on the electrorotation of semiconducting nanowires. The authors state that they observe a new effect whereby the real part of the polarizability is affected by light and that this affects the alignment in a low frequency electric field. Surprising is that light oscillates at much higher frequencies and that the induced moments are far away from the frequencies that are relevant for electrorotation. However, the time averaged polarization times the field square experiences a large enough change that the light can be used to tune the effective polarizability and hence the response to an electric field. The authors demonstrate several applications of nanowire orientation switching and manipulation. The paper focuses on applications, but I find the fundamental effect and its demonstration very interesting, especially if it is true as the authors state that the effect has not been observed before. I would suggest that the authors provide additional information and explain the implications of the change of the real part of the polarizability and that they quantify it. I think that the generality of the phenomenon should be stressed and the light induced polarizability changes should be quantified. I have the following comments for the authors to consider:

1. The results are presented in terms of angles and rotation rates or alignment rates. While this may be of interest in rotating a wire I find these numbers less insightful as they depend on the field strength and the geometry. Can you not please provide some plots of the polarization components (wire itself) and the EDL as a function of frequency.
2. Related to the previous question: Can you provide some idea by how much the polarizabilities are changed when light is present, i.e. can the previous calculations also be shown as a function of the light intensity?
3. Can you rationalize the change in the polarizability with the light intensity. What does that mean with respect to the absorption in a nanowire. Why do you observe the decrease in the rate when the light intensity increase? There should be some physical model that relates photon flux to absorption cross section to induced polarizability.
4. How would this change for other semiconductors?
5. Light will also affect a metal, especially near the plasmon resonance. Would one then not also expect a change in the rotation rates for metallic objects? If the imaginary part dominates, then one could instead consider a dielectric (non-semiconducting) nanowire.
6. Since you relate the change in the real part of the polarizability to the imaginary part and mention the Kramers Kronig relations, I would have expected a calculation that shows that you can indeed relate the newly determined measured real part of the polarizability to the imaginary part with a Kramers Kronig transformation.
7. Is the method you present a new way to characterize the frequency dependent real and imaginary components of the polarizability and in what ways does it complement dielectric spectroscopy?
8. Another effect that the authors have not considered should arise when the authors start to modulate the light at frequencies comparable to the electric field frequencies.

9. „Pullout“ torque seems a strange term. Is this meant to be step-out? Please define the term and perhaps use a different word.
10. For a motor I would expect some forces and torques (holding etc.). Can you provide some numbers (with units)?
11. Can you please give the units of the torque in Fig. S3.
12. Can you specify better in the main part of the paper what the alignment accuracy is that you can potentially achieve.
13. The paper would benefit from editing the English.

Reviewer #2 (Remarks to the Author):

The authors present an experimental and theoretical description of controlled rotation in silicon nanowires as micromotors under a dual stimulus consisting of electric field application (to align the nanowire) and visible light laser exposure (to control the frequency of rotation). As a proof of the accurate control of the nanorotor, the actuation of a single nanowire is used as a communication platform (to send Morse's code messages), and two nearby rotors are independently stimulated to evaluate the decoupled operation. Although, nano/micromotor accurate control is of great interest for interdisciplinary fields, the general experimental and theoretical working principle/concept were already reported by the same group (Science advances, 2018, 4(9), p.eaau0981).

I consider that unless the manuscript novelty is more clearly highlighted, especially versus the group's previous work and other non-electrical nanorotor and tweezer systems, the work should be considered for a more specialized journal.

Specific remarks

1) The novelty should be highlighted in terms of mechanism. What is it new in this method that could not be done when compared to previous literature?

Although this work demonstrates a high degree of control, the combination of both stimuli only shows a small improvement in the nanowire rotation control versus previous literature (e.g., ACS Nano, 2014 9(1), pp.548-554. Science advances, 2018, 4(9), p.eaau0981). Moreover, the presented approach lacks some features common of other electrically actuated microrotors capable of mixing,

transporting cargoes or assembly (e.g., ACS applied materials & interfaces, 2017, 9(7), pp.6144-6152. ACS Nano, 2018, 12(2), pp.1179-1187. Nature materials, 2016 15(10), p.1095.)

2) Nanowires usually tend to aggregate, and in the majority of cases dispersions in water require specific conditions (sonication, specific pH, surfactants, small molecules, etc.) to avoid aggregation, how did you obtain single wires?, how many nanowires per volume do you have in your dispersions? All the experiments are focused on a single nanowire, do all nanowires present a localized behavior when they are stimulated by light and electrical field? How far apart do they have to be from the center of the beam of the laser to not feel the stimuli? How does the response of the nanowire change upon application of the electrical field depending on the position of the nanowire on the platform? Which region in between electrodes is the characterization of the nanowires performed (middle, close to a specific electrode)? Include the reproducibility of the results for a representative number of motors ($n > 10$). Include a video (without blur) in which we could see the collective behavior of several nanowires under the same stimuli.

3) There is a large inconsistency when explaining angles and alignment rates. In the text and some figures, the alignment rate is expressed in units (rad/s). However, the numbers in the graphs and text include from 0 to 500 rad/s, what are the units rad/s or $^\circ/s$ or Hz? What does 500 mean? How were the graphs normalized (include units for those graphs)? Why the normalization is not performed such the LED illumination (white light ~ 500 lux) is the zero? The alignment rate could be defined as the frequency of rotation.

4) The manuscript could be shortened to inform of the most important findings (Morse code messages, independent operation of nearby rotors, programmability) and avoid repetition of previously demonstrated claims. The authors could consider to not include equations and theoretical explanation/ideas already published in previous work in the main text (Science advances, 2018, 4(9), p.eaau0981), as they could be moved to Supplementary Information. Moreover, the presentation of the figures is in some cases similar to the group's recent publications, making it difficult to differentiate/find novelty on the current ones. Therefore, figure redesign could make the manuscript stand out against previous works.

5) The paper makes multiple indirect claims of nanomotor lifetime based on its continuous use, e.g. "light induced alignment switch can be cycled over hundreds of times" page 1 line 13 "which can be cycled for hundreds of times continuously" Page 6 line 103. Could you indicate for how long could a single wire could be used? The authors address potential passivation of the Si-based on the oxidation; therefore, it would be interesting to know what the maximum lifetime of a single nanorotor is before passivation. The same nanowire was tested over how long? Include time scales, eg. every hour, day, etc.,?

6) The scheme in figure 3 illustrates the electrodes as rectangles, they should be represented as trapezoidal shapes for consistency with figure 1.

7) The independent control of two stepper micromotors under the same electric field is quite exciting, however, the concept could be better illustrated by showing the actual microscopy image of the rotation control for both nanowires, rather than of a single nanowire as its currently shown in figure 5B.

Reviewer #1 (Remarks to the Author):

The paper “Light Programmable Micro/Nanomotors with Optically Tunable In-Phase Electric Polarization” by Liang, Teal and Fan describes the effect of light on the electrorotation of semiconducting nanowires. The authors state that they observe a new effect whereby the real part of the polarizability is affected by light and that this affects the alignment in a low frequency electric field. Surprising is that light oscillates at much higher frequencies and that the induced moments are far away from the frequencies that are relevant for electrorotation. However, the time averaged polarization times the field square experiences a large enough change that the light can be used to tune the effective polarizability and hence the response to an electric field. The authors demonstrate several applications of nanowire orientation switching and manipulation. The paper focuses on applications, but I find the fundamental effect and its demonstration very interesting, especially if it is true as the authors state that the effect has not been observed before.

We thank the reviewer for the comments, evaluation, and recognition of our work. We very much agree that the fundamental effect is equally as important as the applications, since it can be generalized to many different materials and several types of nanomanipulation while inspiring applications beyond those which have been demonstrated. Indeed, it is the first time that the effect of optical tunability of the real part polarizability of semiconductor nanoparticles has been observed and reported.

I would suggest that the authors provide additional information and explain the implications of the change of the real part of the polarizability and that they quantify it. I think that the generality of the phenomenon should be stressed and the light induced polarizability changes should be quantified.

Thank you. We have added and discussed additional information as follows. First, implications of changing the real part of the polarizability include refining at least three important manipulation phenomena directly relate to the real part polarization — electro-alignment (studied in this work), dielectrophoresis (transport of particles with electric field gradient), and the chaining effect (particle-particle interactions in electric field). This work lays the foundation for optically controlling of these different types of manipulations, so we think the fundamental effect unveiled here is important (revised manuscript page 4).

We have also included the quantified details of our study of polarizability change with light, and have updated the manuscript (page 9) and Figure. 2a, b, and 3b accordingly.

Generality of the phenomenon: based on our investigation and understanding, such an effect can be generalized to any type of material as long as its electrical conductivity can be changed significantly when stimulated by light or any other stimuli. Semiconductor micro/nanoparticles and molecules made of inorganic or organic materials with tunable electric conductivities under light stimulation, *i.e.* photoconductive materials, may all exhibit such an effect. We demonstrated with the most common type of semiconductor, Si. The specific performance depends on the band structure, carrier lifetime, laser wavelengths, dimension of the particle, and the electrical properties of the suspension medium, and should be investigated on individual materials. In

terms of other types of stimuli, we think that materials that change electric conductivities in magnetic fields, *i.e.* those with high magnetoresistance, could possibly exhibit an analogous effect. (revised manuscript, page 26)

I have the following comments for the authors to consider:

1. The results are presented in terms of angles and rotation rates or alignment rates. While this may be of interest in rotating a wire I find these numbers less insightful as they depend on the field strength and the geometry. Can you not please provide some plots of the polarization components (wire itself) and the EDL as a function of frequency.

Thanks for the helpful comments. We updated the Fig. 2A, B, and 3B with the polarizability shown on the right axis and the alignment rate on the left. We attached the updated figures as follows.

Please find our simulation results showing the electric polarization decomposed into the nanowire's intrinsic polarization and the EDL's contribution; the results are helpful for the understanding of the contributions of these two effects, especially because experimental results

can only show the combined effects. The simulations of Maxwell-Wagner polarizability of a nanowire, EDL polarizability, and combined polarizability in both the longitudinal and transverse directions of a wire are shown in Figure S1 and below. The simulation result of combined effects for the alignment manipulation (the difference of polarizability in the longitudinal direction and transverse direction) is in Fig. 3B.

Figure S1: Polarizability contribution of the Maxwell-Wagner polarization and the electrical double layer. (A, B) Real-part polarizability in the longitudinal direction of a nanowire due to the Maxwell-Wagner polarization and electrical-double-layer effect, respectively. (C, D) Real-part polarizability in the transverse direction of a nanowire due to the Maxwell-Wagner polarization and electrical-double-layer effect, respectively. (E, F) Combined real-part polarizability from both the Maxwell-Wagner and electrical-double-layer effect in the longitudinal and transverse directions of the nanowire, respectively. The combined simulation for the alignment manipulation (the difference of polarizability in the longitudinal direction and transverse direction $\text{Re}(\alpha_{\parallel} - \alpha_{\perp})$) is in Fig. 3B.

2. Related to the previous question: Can you provide some idea by how much the polarizabilities are changed when light is present, i.e. can the previous calculations also be shown as a function of the light intensity?

We included the information in the updated Fig. 2A, B and Fig. 3B with the polarizability shown on the right axis and the alignment rate on the left. In Fig. 3B, the light effect is further exhibited as optoelectric conductivity.

As for the role of light intensity, we included the answer when responding to the following question in which the correlation of electric polarizabilities and light intensity at different AC frequencies has been discussed.

3. Can you rationalize the change in the polarizability with the light intensity. What does that mean with respect to the absorption in a nanowire. Why do you observe the decrease in the rate when the light intensity increases? There should be some physical model that relates photon flux to absorption cross section to induced polarizability.

When laser light (e.g. 532 nm, photon with higher energy than the bandgap of Si) illuminates a nanowire, electrons in the valence band absorb the energy of photons and get excited to the conduction band, leading to an increase of the total number of free electrons and holes. As a result, the electric conductivity of the nanowire increases. In an AC electric field, when the electrical conductivity (σ) increases, the complex permittivity of the nanowire ($\epsilon'' = \epsilon - i\sigma/\omega$, where ϵ , σ , and ω are the real-part permittivity, electric conductivity, and electric-field frequency) will also change, and the total electric polarization in a suspension medium will be changed accordingly.

1) Here we provide a more detailed discussion of the model:

a) Silicon nanowire light absorption:

First of all, for the same nanowire and same wavelength, the absorption cross-section can be considered as a constant within the linear optics regime (intensity of laser is moderate, without non-linear effects). Thus, the light power absorption is proportional to light intensity. Photons absorbed by the Si nanowire will generate pairs of electrons and holes, which drift under external electric fields, resulting in an increase of electrical conductivity and electric dipole moment. The generation rate of electron-hole pairs is proportional to the light intensity within the linear optics regime.

b) Light intensity and electrical conductivity¹:

For a silicon nanowire of 500 nm diameter and 10 μm length, under the illuminance of 318 mW/cm^2 at 532 nm, the total power absorbed is equivalent to the energy of 1.7×10^{10} photons s^{-1} . We adopt 50% as the quantum efficiency for a conservative magnitude estimation, as a result, the photo-excited carrier generation rate (G) is $4.3 \times 10^{21} \text{ s}^{-1} \text{ cm}^{-3}$. When the steady state of the system is reached, there must be a balance established between the photoexcitation and recombination, which indicates the recombination rate of the nanowire must equal to the carrier generation rate (G) of $4.3 \times 10^{21} \text{ s}^{-1} \text{ cm}^{-3}$. The recombination rate of excess carriers can be expressed as $\Delta n/\tau_{eff}$, where Δn is the excess carrier density (for intrinsic silicon, $\Delta n = \Delta p$, and we need to just consider Δn for the purpose of calculation), τ_{eff} is the effective excess

carrier lifetime. For steady state, $G \cdot \tau_{eff} = \Delta n$. The effective lifetime is composed of two parts, the bulk lifetime and the surface lifetime, and can be expressed as $\frac{1}{\tau_{eff}} = \frac{1}{\tau_{bulk}} + \frac{4S}{d}$, where τ_{bulk} is the bulky recombination lifetime mainly relating to Auger, Shockley-Read-Hall and radiative recombination, S is the surface recombination rate and d is the diameter of the nanowire. For nanowires, the bulk recombination is negligible, and the surface recombination dominates. After a comprehensive literature survey, we found that the surface recombination rate of silicon nanowire falls into the magnitude of 10^4 cm s^{-1} without special surface passivation¹⁻⁴, and thus $\tau_{eff} \approx 5 \text{ ns}$ for silicon nanowire of 500 nm diameter. The excess carrier density can then be calculated $\Delta n = G \cdot \tau_{eff} \approx 2.2 \times 10^{13} \text{ cm}^{-3}$. Therefore, we can calculate the photoconductivity given by $\Delta\sigma = (\mu_e + \mu_h)e\Delta n$ with a magnitude of 10^{-1} S m^{-1} .

In short, light with higher intensity will generate more photo-carriers and result in higher electrical conductivity. With reasonable assumptions, the light power $P \propto \Delta\sigma$ can be a good approximation. Thus, in the numerical simulation of the polarizability, we use the electrical conductivity as the sweeping parameter instead of the light intensity for simplicity (only electrical polarization needs to be simulated, greatly simplified the simulation model without calculating optical absorption).

2) Decrease of the alignment rate when light intensity increases:

In experiment, we observed decrease in alignment rate when light intensity increases at low frequencies, *i.e.* 5 kHz, and the simulation result also shows decrease in alignment rate with increase of electrical conductivity in Fig. 3b (<5 kHz). First, as discussed in the manuscript, the alignment rate is directly proportional to the value of $\text{Re}(\alpha_{\parallel} - \alpha_{\perp})$, at low frequencies, $\alpha_{\perp} \ll \alpha_{\parallel}$, for simplicity, we only consider the polarization in the longitudinal direction of the nanowire (α_{\parallel}). As shown in Figure S2 (also as follows). we plot both the real and imaginary parts of the Maxwell-Wagner polarizability of the nanowire. If we model the nanowire made of pure silicon without surface oxidation, then the results are shown in Fig. S2b, d. The real part of the polarizability increases as the light intensity increases, and the imaginary part of the polarizability is almost zero at low frequencies. However, once we include a thin oxide layer on the surface of the Si in the model, since Si always has a native oxidation layer, the corresponding simulation result is shown in Fig. S2a demonstrates a decrease in the real-part polarizability at low frequency (<10 kHz). Meanwhile, the imaginary polarizability increases at low frequencies (Fig. S2c). After considering the effect of electrical double layer, the final results show decrease in real polarizability as electrical conductivity increases.

In a more intuitive picture, the thin oxide layer can be regarded as an additional capacitor standing in between the silicon nanowire and the electrical double layer. The dipole moment generated on the silicon nanowire first induces the polarization of the oxide capacitor, and then induces the EDL polarization. However, the polarization of oxide layer causes additional phase lag, as indicated in Fig. S2. The norm of the total polarization

$(\sqrt{\text{Re}(p)^2 + \text{Im}(p)^2})$ still increases as the light intensity increases, but due to the existence of the oxide capacitance and the resulting additional phase lag, the imaginary part of the polarizability increases greatly, while the real part slightly decreases.

Figure S2: Consideration of surface oxidation of Si nanowires on the Maxwell-Wagner polarization. (A) The simulation results of real-part polarizability from Maxwell-Wagner polarization with and (B) without the consideration of a thin oxidation layer on the silicon nanowire surface. (C) The simulation results of imaginary part polarizability from Maxwell-Wagner polarization with and (D) without the consideration of a thin oxide layer on the silicon nanowire surface.

4. How would this change for other semiconductors?

Silicon is a widely used semiconductor with many applications. That is why we utilized it to study the effect. Although we haven't tested other semiconductors, in principle, as long as the materials' electric conductivity can be modulated by light with prominent changes, such an effect should be observable. In particular, as silicon is an indirect bandgap semiconductor, which has relatively low absorption of light, the same effect could be expected with an even higher

efficiency from direct bandgap semiconductors *i.e.* GaAs. It is possible that semiconductor 2D materials and photoconductive polymers exhibit the same effect; this would need further investigation.

5. Light will also affect a metal, especially near the plasmon resonance. Would one then not also expect a change in the rotation rates for metallic objects? If the imaginary part dominates, then one could instead consider a dielectric (non-semiconducting) nanowire.

Experimentally, we have tested the light effect (532 nm) on alignment of gold nanowires (200 nm diameter and 5 μm in length), however, no change in alignment rate has been observed. Although it is not in plasmonic resonance, we do not think we can observe the same effect on metallic nanowires even there is a plasmonic resonance due to the following reasons: First, when a metal wire is exposed to a laser the energy of photons is absorbed by free electrons, which is different from the absorption process in semiconductor materials. If the laser wavelength is close to the plasmon resonance wavelength of the nanowire, the absorption cross section can greatly increase, causing strong free electron oscillation at same frequency of light (optical frequency $\sim 10^{14}$ Hz) and strong electric field near the surface. The electric field induced rotation is operated under AC electric field at much lower frequencies *i.e.* below 10 MHz, the plasmon resonance is at least 7 orders of magnitude higher than the maximum AC frequency that we used to actuate the electrorotation, which means that in one AC E-field oscillation cycle, the polarization of metal induced by light will go through millions cycles of oscillation, and cancels out in average. Thus, we do not think such effect can be observed in metallic nanowires.

Here we want to emphasize that the frequency of light as stimulation and the frequency of the AC electric field that drives the rotation should be considered separately. The light frequency ω_1 governs the complex permittivity $\epsilon'(\omega_1) + i\epsilon''(\omega_1)$ and the corresponding optical properties (absorption). The AC electric field frequency ω_2 governs the complex permittivity at AC frequency $\epsilon'(\omega_2) + i\epsilon''(\omega_2)$, which determines the electrical polarization and the corresponding electric-field-induced mechanical motions (alignment, rotation).

As for the electro-rotation, in which the rotational speed is governed by the imaginary part, there must be certain types of relaxation processes to cause non-zero imaginary part, *i.e.* interface relaxation (Maxwell-Wagner), and electrical double layer (ionic transportation). As for dielectric non-semiconductor materials, usually, they are very insulating with much lower polarizabilities (in both real and imaginary parts), and since they do not respond to light with change of electric conductivity, we do not think they will exhibit the reported effect. However, for the photoconductive polymers, they may have similar effect which requires further study.

6. Since you relate the change in the real part of the polarizability to the imaginary part and mention the Kramers Kronig relations, I would have expected a calculation that shows that you can indeed relate the newly determined measured real part of the polarizability to the imaginary part with a Kramers-Kronig transformation.

Thanks for the very good question. We observed the light induced change in the imaginary part of polarizability as shown in our previous study. From what the K-K relation suggests, the real

part of the polarizability could also be changed by light. Based on this, we find it will be interesting to experimentally study the effect of light on the real part of polarization of semiconductor nanowires.

However, at this stage, we are not able to directly calculate the real part imaginary part of polarizability from the previously experimentally measured imaginary part because of the following obstacles:

First, in our previous work, we measured the imaginary part of nanowire polarizability from the experiment of electro-rotation. The electro-rotation experiment is conducted by applying a circularly polarized AC electric field (direction of the field is rotating at the same frequency of oscillation). The electro-rotation speed is directly proportional to $\text{Im}(\alpha_{\parallel} + \alpha_{\perp})$. However, in this work, the electro-alignment experiment is based on the real-part of polarization, in which the alignment rate is proportional to $\text{Re}(\alpha_{\parallel} - \alpha_{\perp})$. According to Kramers-Kronig relation, $\text{Im}(\alpha_{\parallel})$ and $\text{Re}(\alpha_{\parallel})$ are correlated, and $\text{Im}(\alpha_{\perp})$ and $\text{Re}(\alpha_{\perp})$ are correlated. Although α_{\perp} is usually much smaller compared to α_{\parallel} (Fig. S1A, C), we are not able to strictly determine $\text{Re}(\alpha_{\parallel} - \alpha_{\perp})$ from the experimentally obtained $\text{Im}(\alpha_{\parallel} + \alpha_{\perp})$ via the K-K relation.

Here, we still mentioned the suggestion of the K-K relation because, for a nanowire with a long aspect ratio, the polarizability along the longitudinal axis is usually much larger than that along the transverse axis, i.e. $\text{Im}(\alpha_{\parallel}) \gg \text{Im}(\alpha_{\perp})$ and $\text{Re}(\alpha_{\parallel}) \gg \text{Re}(\alpha_{\perp})$, which are satisfied at most frequencies (not including specific frequencies when $\text{Im}(\alpha_{\parallel})$ and $\text{Re}(\alpha_{\parallel})$ are close to zero, e.g. as that shown in Fig. S1 A and C). So we think the K-K relation could suggest the light tunability of the real-part of polarizability from the experimental observed light tunable imaginary-part of polarizability.

Another more important obstacle comes from the experimental aspect. To implement the K-K relation, the integral contour is the entire upper half-plane, which means, the value of one part, either imaginary or real, has to be known over all frequencies. Experimentally, only a limited range of frequencies can be measured, and the common way to implement the K-K relation is to interpolate or extrapolate for the missing data via certain approximation. For our case, at high frequency limit, both real and imaginary parts of polarizability approach zero, however, at low frequencies *i.e.* below 1 kHz, the behavior of polarizability remains unmeasurable because of the co-existence of other electrokinetic effects: first, at low frequencies, the electrical double layer at the electrode-solution interface will screen most voltage for effective electric field applied on a nanowire; second, at low frequencies, AC electric field will induce strong AC osmosis flow in aqueous solution; last, electrohydrolysis reaction starts at lower voltages with reduced AC frequency. All of them conceal the motion of nanowire from the polarization effect. Due to the above, we are not able to measure the electric polarization generated motions at low frequencies to quantitatively determine the real-part and imaginary parts of polarizability from each other by using the K-K relation.

(The above mentioned obstacles may be solved by changing the solution to a non-aqueous medium so that the electrokinetic side effects can be suppressed.)

7. Is the method you present a new way to characterize the frequency dependent real and imaginary components of the polarizability and in what ways does it complement dielectric spectroscopy?

The method of electro-rotation (our previous work, circularly polarized rotating AC field) has been developed since the 1980s ([https://doi.org/10.1016/0304-3886\(88\)90027-7](https://doi.org/10.1016/0304-3886(88)90027-7)). In early stage, this technique is most attractive in its application to characterize the dielectric properties of cell membrane structures (multilayer structure has multiple dielectric dispersion peaks).

The method presented in this paper can potentially be used to determine the real part of polarization of individual micro/nanoparticles with some limitations. Since the alignment rate is determined by $\text{Re}(\alpha_{\parallel} - \alpha_{\perp})$, the particle has to be asymmetric in shape. A spherical particle will not experience alignment torque under the AC electric field. Furthermore, unless the particle has a long aspect ratio, satisfying $\alpha_{\parallel} \gg \alpha_{\perp}$, the measurement can only obtain the real part of the difference of the polarizability along two axes.

For the application of electro-rotation and electro-alignment as a method of dielectric spectrum, in addition to the longitudinal shape anisotropy of micro/nanoparticles, we need to pay attention to the use of suspension medium. Since several electrokinetic effects exist in aqueous solution under electric fields as aforementioned, which make it difficult to extract the information of dielectric response of particles from experimental data. It will be desirable to suspend particles in non-polarized medium, such as oil. Then, this method could be used for the dielectric spectrum of longitudinal particles.

The advantages of this method compared to traditional dielectric spectroscopy is also clear. 1) It could be used for determining the dielectric spectra of individual micro/nanoparticles; 2) it is non-invasive; 3) no requirement of direct electric contact.

8. Another effect that the authors have not considered should arise when the authors start to modulate the light at frequencies comparable to the electric field frequencies.

The time-scale of photon-electron interaction is ~ 10 fs, and the following electron-phonon, electron-electron interaction relaxation processes towards a steady state are in the range of picoseconds. These are all ultrafast processes, much faster than the camera frame rate of milliseconds. Therefore, we can consider the response of electro-rotation to optical stimulus as instant. A sudden removal of laser stimulus after a continuous exposure will be followed by the relaxation process, which depends on the decay rate or lifetime of the excess carriers. According to previous reported experimental measurements, usually the lifetime of excess carriers in bulk Si is tens of microseconds (low doping), yet for silicon nanowires, the lifetime is even shorter due to the high density of surface states and the corresponding high rate of recombination. Therefore, the relaxation process is also instant compared to that of the camera frame rate. The optical responses are very fast for both off-to-on and on-to-off processes.

As asked by the reviewer, it can be interesting if we modulate light at the comparable frequencies as the AC electric field. Though our current setup is not able to modulate light at speeds higher than 30 Hz, and thus we cannot test experimentally, it is still interesting to think of the question. Here we provide some basic thoughts. For simplicity, we assume that the light modulation frequency is same as the AC electric field frequency. Then, there will be two simple scenarios, first, when the AC frequency is much higher than the relaxation frequency of the carriers, the time interval between two consecutive light pulses is not long enough for the photon-generated carriers to be depleted due to recombination, and the photon-generated carrier density would maintain a certain level in average, and thus the effect would still exist. If the AC frequency is much lower than the relaxation frequency of the carriers, the phase between light stimulation and AC electric field starts to play an important role. If the light is on when the AC E-field reaches the maximum amplitudes, then the effect should still exist. If the light is on when the E-field amplitude is close to zero, then the effect should be weak. Overall, it is very interesting to think about modulate light at similar frequencies as the electric field, finer control of the polarizability could be achieved.

9. „Pullout “torque seems a strange term. Is this meant to be step-out? Please define the term and perhaps use a different word.

Pullout torque of a stepper motor is the highest torque a stepper (or any synchronous) motor can provide to an external load at a given speed without losing steps. If the motor has a load requiring a higher torque than this, it loses synchronous step motions.

10. For a motor I would expect some forces and torques (holding etc.). Can you provide some numbers (with units)?

Thank you. We have updated the figure S5 (also attached as following) to show the output power and torque in units of W and $N \cdot m$, respectively, instead of the normalized unitless torque.

11. Can you please give the units of the torque in Fig. S3.

Thank you. The units of torque have been updated.

12. Can you specify better in the main part of the paper what the alignment accuracy is that you can potentially achieve.

We have included Fig. 5C with a more detailed analysis of alignment accuracy. Our step motor can provide a restoring force of $\tau_{\text{res}} = A(\omega, E_0) \sin \theta \cos \theta$, in which the alignment rate A is a function of AC frequency and the electric field strength. It is obvious that the alignment accuracy increases as the electric field strength increases or if the selected frequency has stronger polarizability. The updated Fig. 5C shows the angle fluctuation of a single nanowire from the designated alignment angle at 100 kHz E-field and different applied voltages ($\propto E_0$). As the voltage increases, the angle distribution becomes narrower with smaller standard deviation (σ) fit by a normal distribution.

13. The paper would benefit from editing the English.

We will carefully proofread the manuscript again. Thank you!

Reviewer #2 (Remarks to the Author):

The authors present an experimental and theoretical description of controlled rotation in silicon nanowires as micromotors under a dual stimulus consisting of electric field application (to align the nanowire) and visible light laser exposure (to control the frequency of rotation). As a proof of the accurate control of the nanorotor, the actuation of a single nanowire is used as a communication platform (to send Morse's code messages), and two nearby rotors are independently stimulated to evaluate the decoupled operation. Although, nano/micromotor accurate control is of great interest for interdisciplinary fields, the general experimental and theoretical working principle/concept were already reported by the same group (Science advances, 2018, 4(9), p.eaau0981).

I consider that unless the manuscript novelty is more clearly highlighted, especially versus the group's previous work and other non-electrical nanorotor and tweezer systems, the work should be considered for a more specialized journal.

Thank you for the comments. However, we think there are misunderstandings of this report with respect to our previous work (cited in ref. 31 in the original manuscript). Here we want to emphasize the novelty of this work from two aspects: the importance as a fundamental study of the physics underlying the reported phenomenon, and the unprecedented applications that cannot be achieved with the working mechanism reported in previous works. The interest of this new fundamental study with demonstrated applications was also commented by the first reviewer. We confirm what we have reported is the first observation and study of the light tunable real-part polarization. Here, to clarify the misunderstandings, we have elaborated the novelty in the context of ref. 31 and other types of manipulations. Please find the discussion in the revised

manuscript on page 3-4, 20, 27-28, which is also included in the answer to the specific remark (1) as follows.

Specific remarks

1) The novelty should be highlighted in terms of mechanism. What is it new in this method that could not be done when compared to previous literature?

The novelties of this work are exhibited in both theoretical and experimental aspects:

First, the fundamental investigation of the optically tunable real-part (or in-phase) polarization of semiconductor nanowires in an electric field is a new fundamental research. As commented by the first reviewer, the fundamental study is of paramount interest as the demonstrated applications. This is a new fundamental study because of the following:

(1) Our previous study (Science Adv. 2018) unveiled the light-effect on the imaginary-part (out-of-phase) polarization of nanowires with theoretical analysis and the electro-rotation experiments, where nanowires are driven to rotate in an external electric field rotating at a much higher frequency, in which the out-of-phase electric interaction compels the motions of nanowires. However, from the obtained theoretical and experimental results, one *cannot* predict, derive, or understand that light can tune the real-part polarization of nanowires to generate the corresponding real-part-polarization governed manipulations, as shown in this study. Here, the real-part electric interaction governed manipulations include multiple important and distinct types of motions that *cannot* be achieved based on that reported previously (the imaginary-part-electric interactions), including electric alignment (demonstrated in this work), transport of wires from one location to another (dielectrophoresis, in investigation), and chaining among a swarm of nanowires (in preparation). *Fundamentally, none of the above manipulations can be predicted, explained, or guided by utilizing knowledge obtained in our previous work.* We will further explain this in the following:

According to the theoretical Kramers-Kronig relation given by the following equations:

$$\chi' = -\frac{1}{\pi}P \int_{-\infty}^{\infty} \frac{\chi''(\omega')}{\omega' - \omega} d\omega'$$

$$\chi'' = \frac{1}{\pi}P \int_{-\infty}^{\infty} \frac{\chi'(\omega')}{\omega' - \omega} d\omega'$$

Here χ' and χ'' are the real and imaginary part of susceptibility, respectively, P is the Cauchy principle value of the integral.

The real and imaginary parts of polarization of a material in an electromagnetic field are correlated, changes in the imaginary part should cause change in the real-part polarization. However, to implement the K-K relation, one should carry out the integration on the entire upper half-plane, which means, the value of one part, either imaginary or real, has to be known over all frequencies to determine the other part. Our previous experimental study on the imaginary-part polarization based on the out-of-phase electrorotation *cannot* be utilized to predict the optical effect on the real-part polarization, since experimentally, only a limited range of frequencies can

be measured. The common way to implement the K-K relation is to interpolate or extrapolate for the missing data via certain approximation. In our case, at high frequency limit, both real and imaginary parts of polarizability approach zero, however, at low frequencies *i.e.* below 1 kHz, the behavior of polarizability of nanoparticles in suspension is unmeasurable or very difficult to measure because of the co-existence of other electrokinetic effects: first, at low frequencies, the electrical double layer at the electrode-solution interface screens most voltage for effective electric field applied on a nanowire; second, at low frequencies, AC electric field induces strong AC electroosmosis flows in aqueous solution; last, hydrolysis reaction starts at lower voltages, e.g. 3V, at low AC frequency. All of them conceal the motion of nanowire originated from the polarization effect. Due to the above, the availability of the K-K relation may suggest the possibility of light-tunable effect on the in-phase polarization given our previous finding (ref. 31.) However, it *cannot* predict, not to mention to understand any light effect on the real-part polarization of nanowires, as reported in this work; *these must be found experimentally*. There is why we consider the work reported here is very important as a fundamental study.

In the experimental and application aspects, the investigation of optical effect on the real-part polarization in this work lays the foundation for several distinct and important types of manipulations, as mentioned earlier, which cannot be obtained with our previous work, including the alignment (this work), transport (dielectrophoresis), and chaining of particles. For instance, the demonstrated (i) optical switchable alignment for Morse signaling and (ii) optical controlled step motors cannot be achieved with that reported previously (tunable imaginary polarization), of which the only important phenomenon is the electro-rotation based on the out-of-phase electric polarization where a wire rotates at a much lower frequency than that of the electric field. From the above discussion, one can find that the types of manipulations and potential applications based on what has been reported here, the optically tunable real-part polarization, are much greater compared to that studied previously.

In terms of other non-electrical manipulation methods, there are several works that combine two different types of forces to generate responsive motions based on the respective working mechanisms, e.g. magnetic-catalytic and magnetic-acoustic motors. However, it is a rarely exploited concept to generate switchable mechanical operations of nanomotors via instantaneous tuning of their physical properties, e.g. light tunable electrical conductivity.

Therefore, we consider this work is sufficiently novel in the regards of fundamental study, demonstrated applications, and potential impact for this journal.

Although this work demonstrates a high degree of control, the combination of both stimuli only shows a small improvement in the nanowire rotation control versus previous literature (e.g., ACS Nano, 2014 9(1), pp.548-554. Science advances, 2018, 4(9), p.eaau0981).

We respectfully disagree with this comment. In the following we will discuss to clarify the misunderstanding.

First of all, both of the above cited works are based on electro-rotation, resulted from the out-of-phase (imaginary-part) electric polarization. The working mechanism is entirely different from what has been reported in this work based on the in-phase polarization. With the new manipulation mechanism based on the in-phase (real-part) polarization, the created stepper motor

in this report exhibits at least three distinct advantages compared to that in previous works: (1) the time-dependent angle of a motor can be completely and precisely controlled by the applied E-field, since a motor rotates with the E-field synchronously, without the use of any imaging feedback as that in our previous study. (2) Under proper lighting conditions, all the motors, regardless of their differences in geometry or electrical properties, can be aligned in the same direction synchronously, which makes it possible to control arrays of motors to rotate at an exact same speed under the same field (as shown in **Fig. 5**). This was *not* achieved in our previous study, since differences in each motor result in distinct torques and speeds, not synchronous with the electric field. (3) The alignment field in this work can provide an effective control torque to fix the motor's angle when it is stopped, countering the Brownian motion as shown in **Fig. 5C**, which was not demonstrated previously. Furthermore, the motors reported in ACS Nano, 2014 9(1), pp.548-554 *cannot* be modulated by light.

Overall, the stepper motor reported here has clear distinctions and advantages compared to previous motors.

Moreover, the presented approach lacks some features common of other electrically actuated microrotors capable of mixing, transporting cargoes or assembly (e.g., ACS applied materials & interfaces, 2017, 9(7), pp.6144-6152. ACS Nano, 2018, 12(2), pp.1179-1187. Nature materials, 2016 15(10), p.1095.)

We respectfully disagree with this evaluation. In this work, we have already studied new physics with experiments and modeling, and demonstrated two unprecedented applications based on the effect. The applications that have been demonstrated previously, including those by ourselves *e.g.* mixing, can be surely realized, but there will not be much novelty and we cannot report all applications in a single paper. Instead, we write this paper in a way to only focus on the new fundamental physics with applications only achievable by the investigated effect, *i.e.* the unprecedented applications of Morse-code signaling and individually controlled stepper motors.

Here, we also expect this work to make impact on other applications, e.g. cargo delivery and assembly mentioned by the reviewer, which are governed by the real-part polarization enable manipulations, e.g. cargo delivery based on transport (dielectrophoresis, we have observed the effect in experiments and are investigating it) and assembling (chaining in electric fields) of micro/nanoparticles. Again, the physics unveiled in this work has laid the foundation for these important manipulations, however, given the content and length of this paper, we do not think it appropriate to include everything into a single paper. It will make it difficult for readers and unacceptable as an article (word limit of 5000) for Nature Communications.

2) Nanowires usually tend to aggregate, and in the majority of cases dispersions in water require specific conditions (sonication, specific pH, surfactants, small molecules, etc.) to avoid aggregation, how did you obtain single wires?, how many nanowires per volume do you have in your dispersions? All the experiments are focused on a single nanowire, do all nanowires present a localize behavior when they are stimulated by light and electrical field? How far apart do they have to be from the center of the beam of the laser to not feel the stimuli? How does the response of the nanowire change upon application of the electrical field depending on the position of the nanowire on the platform? Which region in between electrodes is the characterization of the

nanowires performed (middle, close to a specific electrode)? Include the reproducibility of the results for a representative number of motors ($n > 10$). Include a video (without blur) in which we could see the collective behavior of several nanowires under the same stimuli.

- 1) Silicon nanowires will naturally form a thin oxidation layer once in air. The oxidized layer forms a negatively charged surface in DI water. Therefore, between nanowires, the electrostatic force is repulsive, which ensures a good dispersion of nanowires. We always sonicate the wires for 5 minutes before carrying out experiments. For single nanowire experiment, we usually dilute the solution to a concentration of ~ 10 nanowires/ μl .
- 2) The reason we studied single nanowire is to exclude the interferences from neighboring wires so that we can obtain reliable experimental results that can accurately reflect the physics we are studying. When two wires are close to each other, additional interaction will be exerted onto the wires, and such local interaction can disturb the rotation or alignment originated from the external electric field, which is what we studied. The interaction between two wires is called chaining effect, which originates from the interaction between the induced electric dipole of each wires. Chaining effect results in attractive interaction between two neighboring wires, with a force inversely proportional to r^4 (dipole-dipole interaction).
- 3) The minimum distance between a nanowire and the center of laser spot so that wire will not be stimulated by light depends on the length of the wire, resolution of digital light processing system, as well as the diffraction limit of the light. The length of wires used in experiments are round $10 \mu\text{m}$. The minimum laser spot size we can realize with the current setup (DMD: TI DLP7000) is $10 \mu\text{m}$, which is much higher than the diffraction limit of 532 nm laser (potentially can be further improved). Thus, the size of the wire and the resolution of the Digital Light Projection (DLP) system determine the minimum distance between the light and a wire so that the wire does not feel the stimuli. The value can be estimated as half length of the wire plus the radius of the light spot, which is $10 \mu\text{m}$.
To ensure that two wires can be independently controlled by light, for now, we can conservatively estimate the minimum distance as two times of the distance that wire does not feel the light, which is $20 \mu\text{m}$. With further improvement in light projection resolution, the distance could be further reduced to sub $10 \mu\text{m}$, at which the electrical interactions between two wires should be considered as a dominating factor (chaining effect).
- 4) We only characterized those nanowires located in the center area of the quadruple microelectrodes (a square area of $\sim 100 \mu\text{m} \times 100 \mu\text{m}$ in the center). We did not study wires that were off the center of the microelectrodes. In this way, a well uniform and reproducible electric-field can be ensured, as shown in the simulation in the follows and Fig. S8. The lines are electric fields, and color scale indicates the electrical potential distribution in the area between the electrodes. Here 0 V potential is applied on electrode 1 and 2, 30 V potential is applied on electrodes 3 and 4. In the center, a fairly uniform electric field orienting at 45° is generated. Off the center, the electric field is much less uniform.

Figure S8: Simulation of electric field distribution in a quadruple microelectrode. Electric field lines show the uniformity of the field.

- 5) To prove the reproducibility of the results of motors performance, we here provide results of 9 more motors in addition to the one that already been shown in Fig. 4C in the manuscript.

Figure S6: Light tunable in-phase and out-phase rotation of a Si nanowire stepper motor. Average rotation speed of a micromotor versus driving voltage with (in blue) and without laser illumination (in orange). The driving electric field is 1.49 Hz, same as that of the in-phase rotation speed of the nanowire.

6) As advised, we included a video showing the rotation behaviors of multiple nanowires (>10) under a same stimulus (supplementary video 11).

3) There is a large inconsistency when explaining angles and alignment rates. In the text and some figures, the alignment rate is expressed in units (rad/s). However, the numbers in the graphs and text include from 0 to 500 rad/s, what are Are the units rad/s or %/s or Hz? What does 500 mean?

Alignment rate and angle: the alignment rate (A) is used to characterize the speed at which nanowire oriented to the electric-field direction, it has the same unit as that of angular speed, which is radian per second (rad/s). The alignment rate (A) is only used to characterize the alignment experiment, in which parallel electrodes are used, nanowires do not continuous rotate, but only align to and stop at the electric-field direction (or perpendicular to electric field). The alignment rate (A) has the value of twice of the angular speed when the angle between the

nanowire and the E-field is $\theta = 45^\circ$. As shown in equation (3) in the manuscript, the equation of motion for the alignment is given by $\dot{\theta} = A \sin \theta \cos \theta$, and the angular speed is $\dot{\theta} = \frac{A}{2}$ when $\theta = 45^\circ$. For instance, if $A=500$ rad/s (as asked by the reviewer), it means that the nanowire's instantaneous rotation speed is 250 rad/s when $\theta = 45^\circ$.

As for the motor rotation experiment (second part of the paper, related to step motor applications), the nanomotors continuously rotate to follow the direction of the rotating electric field. The rotational speed ω has the same unit with that of alignment rate (rad/s), and the rotational speed can also be converted to unit of Hz (or rps), for example, $\omega = 10$ rad/s = $\frac{10}{2\pi}$ Hz or rps (revolution per second), all these units are commonly used for rotation speed.

In summary, the alignment rate is measured in the electro-alignment experiments in parallel microelectrodes. Nanowires are oriented toward and stop at the electric-field direction without continuous rotation. Thus, there is no frequency of rotation involved. As for the stepper nanomotor, it continuously rotates synchronously with the driving AC electric field at a constant speed. The motor's rotation frequency is controlled by the rotation speed (ω_2) of the AC electric-field oscillating at higher frequency (ω_1). The alignment experiments are used to understand the physics. The continuously rotating nanomotors are obtained with such understanding. They are two different experiments.

How were the graphs normalized (include units for those graphs)?

To avoid confusion, we updated Fig. 2B, and Fig. 3B without the use of normalization. All figures now are showing the exact values of alignment rate and polarizability with unit.

Why the normalization is not performed such the LED illumination (white light ~500 lux) is the zero?

We are not able to capture an optical image without the minimum LED illumination; thus, we cannot normalize with LED illumination is zero. However, the influence of the LED illumination is negligible, because the light intensity from LED, ~ 500 lux, can be converted to ~ 0.07 mW/cm², which is more than two magnitudes lower compared to the intensity of laser. Furthermore, we updated the figures with the absolutely values without normalization in Fig. 2A,B and Fig. 3B as mentioned above.

The alignment rate could be defined as the frequency of rotation.

Thank you for the comment. As previously discussed, the alignment rate has the same unit of angular speed (rad/s), which could also be converted it to Hz or rps (after dividing by 2π). However, since alignment experiment does not involve continuous rotation, we think the unit of rad/s could be better for characterizing this phenomenon.

4) The manuscript could be shortened to aid inform of the most important findings (Morse code messages, independent operation of nearby rotors, programmability) and avoid repetition of previously demonstrated claims. The authors could consider to not include equations and

theoretical explanation/ideas already published in previous work in the main text (Science advances, 2018, 4(9), p.eaau0981), as they could be moved to Supplementary Information. Moreover, the presentation of the figures is in some cases similar to the group's recent publications, making it difficult to differentiate/find novelty on the current ones. Therefore, figure redesign could make the manuscript stand out against previous works.

We appreciate the comment sincerely. However, as aforementioned, we think there are some misunderstandings regarding the theoretical and modeling part of this paper compared to that of our previous paper. The distinctions have been discussed in the response to question 1. Since the physics studied in this paper is different from that reported previously, although some of the elementary equations have similarities, we respectfully believe it is important to keep the fundamental study in the main text. Also, we have tried our best to be as concise as possible.

Here, we included some elementary equations, e.g. electrical polarizability, rotational drag force on a nanowire in low-Reynolds-number fluid, and the electrical double layer induced dipole, same as those in previous work, so that the overall physics picture can be complete and understandable.

Thank you. We updated the figures to make them more stylish.

5) The paper makes multiple indirect claims of nanomotor lifetime based on its continuous use, e.g. "light induced alignment switch can be cycled over hundreds of times" page 1 line 13 "which can be cycled for hundreds of times continuously" Page 6 line 103. Could you indicate for how long could a single wire could be used? The authors address potential passivation of the Si-based on the oxidation; therefore, it would be interesting to know what the maximum lifetime of a single nanorotor is before passivation. The same nanowire was tested over how long? Include time scales, eg. every hour, day, etc.,?

Thank you for the question. The statement of "light induced alignment switch can be cycled over hundreds of times" is referring to the light induced switching between parallel alignment to transverse alignment. In Figure 2C, we only show the data for over 60 cycles, and the complete data set we measured include more than hundreds of cycles. The above statement is meant to confirm the repeatability of such an effect, not implying the lifetime of motors.

We carried out the lifetime study of such motors by keeping tracking the change of switching voltages above which a motor switch from out-phase oscillation to in-phase synchronous rotation using a same light (318 mW/cm^2 532-nm laser). Here, please find the following figure showing the required switching voltages over 8 days (SI: Fig. S7). With time, the switching voltage gradually increases from 17 to 23 V, which indicates that the motor's optical response and electrical polarizability decay with time, which is expected due to the oxidation effect in water suspension. After day 8, our set up with a maximum voltage of 30 V cannot reach sufficient voltages for the measurement. In terms of device application, the lifetime of the motor is about 1 week.

Here, we would like to mention that all our experimental results were obtained from motors freshly made within 2 days for reliability. The device lifetime could be improved by surface coating to prevent the oxidation.

Figure S7: Switching voltages required by Si micromotors stored in DI water from day 1 to day 8.

6) The scheme in figure 3 illustrates the electrodes as rectangles, they should be represented as trapezoidal shapes for consistency with figure 1.

Thank you for the comment. We corrected the shape of the electrodes in Figure 3. However, in the simulation, it is the parallel electrode being modelled to generate the uniform electric field and induce polarization instead of the quadruple electrode. The previous diagram with four electrodes was misleading, and we already made the correction and updated Fig. 3a.

7) The independent control of two stepper micromotors under the same electric field is quite exciting, however, the concept could be better illustrated by showing the actual microscopy image of the rotation control for both nanowires, rather than of a single nanowire as its currently shown in figure 5B.

Thanks for the comment. We updated Fig. 5A to show the independent control of the rotation modes of the two motors.

We are sorry that there could be misunderstanding of Fig. 5B. Fig. 5B is a demonstration of another function of our step motor, the angular control of a single stepper motor, where the motor can be aligned to an arbitrary angle. We have also done additional experiments to characterize the accuracy of angular control as follows and **Fig. 5C** in the manuscript.

Figure 5C: The angle fluctuation around the designated angle under different voltages. Probability density is fitted by normal distribution.

References:

- 1 Dan, Y. *et al.* Dramatic Reduction of Surface Recombination by in Situ Surface Passivation of Silicon Nanowires. *Nano Letters* **11**, 2527-2532, doi:10.1021/nl201179n (2011).
- 2 Demichel, O. *et al.* Surface recombination velocity measurements of efficiently passivated gold-catalyzed silicon nanowires by a new optical method. *Nano letters* **10**, 2323-2329 (2010).
- 3 Grumstrup, E. M. *et al.* Ultrafast carrier dynamics in individual silicon nanowires: Characterization of diameter-dependent carrier lifetime and surface recombination with pump-probe microscopy. *The Journal of Physical Chemistry C* **118**, 8634-8640 (2014).
- 4 Kato, S., Yamazaki, T., Kurokawa, Y., Miyajima, S. & Konagai, M. Influence of fabrication processes and annealing treatment on the minority carrier lifetime of silicon nanowire films. *Nanoscale research letters* **12**, 242 (2017).

Dear Dr. Barbera,

Thank you for the decision to accept our manuscript after the final revision. We also thank both reviewers for their valuable and helpful comments and the support of our work. Please find our point-by-point responses as follows:

REVIEWERS' COMMENTS:

Reviewer #1 (Remarks to the Author):

The paper describes the influence of a light beam on the real part of the polarizability (strictly, the anisotropic part of the polarizability). The effect provides a new handle in the manipulation of nanowires: rotation, orientation and switching of direction. The effect is interesting and the authors have given detailed responses to my questions and have added new and useful information. Overall, I think the results are, as far as I can tell, novel and of interest to the community. I am pleased that more detailed discussions have been added about the origin of the effect and that the numerical estimates agree with known properties of nanowires. A small difference between ‘theory’ and experiment is that the simulation predicts an increase in whereas the observed effect shows a decrease of the effective anisotropic polarizability (Fig 3b, inset). This means that the model does not capture all the effects in a frequency range, but there are many and the effects especially when it relates to ion transport near surfaces are complex -- and the authors provide suggestions along these lines. The phenomenology is rich and so that I am sure further insight and details will emerge in future. The main experimental results are convincingly explained and match with numerical estimates and simulations. In my opinion, there are plenty of interesting results that the community can now judge the effect and its usefulness and build on this work. I find the greatly expanded SI material very helpful for the reader. I think the main message is that light can be used to influence the wires electrical properties which greatly expands the scope of electro-rotation, alignment and manipulation of semiconducting nanowires and in turn gives rise to many interesting effects that have been nicely demonstrated.

We thank the reviewer for the insightful evaluation and positive recognition of our work. We deeply appreciate the quality of the review. Please find our responses to the questions as follows:

I have some minor comments and suggestions for the authors to consider:

1. Does Figure 1C suggest that the experiment is conducted in a vertical geometry?

Thanks for pointing out the potentially misleading information from Fig. 1C. The schematic of the experimental setup in Fig. 1C was prepared in a way that the relative positions of all components can be clearly displayed. In the actual experiment setup, the microelectrodes face upward. In the revision, we included this information in the caption of the figure.

2. What limits the number times the effect can be repeated. The authors write on page “which can be cycled for hundreds of times continuously” – is this a limit? In the SI they also add something on days that the measurements can be used, but does the statement say the wires stop

to work when the experiment is repeated 1000 times? If this is the case, then this warrants an explanation. If not please make this clear in the description.

Thanks for the comments. The number of switching cycles demonstrated by us, *i.e.* a few hundreds of times, is not the limit that our nanowire can be used as a switchable device. The wires can keep switching as long as they maintain their positions in the microelectrodes. However, we found that the duration of the nanowires stored in water can result in degraded performance, as shown in the supplementary note 10. The optical response of the nanowires gradually decreases in a week. To improve the lifetime of the nanowires, in the discussion, we suggested to passivate their surfaces.

To make this clear, we changed the description as follows: “the alignment torques on silicon nanowires remarkably change, resulting in accelerated alignment speed and even switching between the parallel and perpendicular directions. Here, we have successfully switched the alignment a Si nanowire between the two directions for hundreds of times, which, however, is not the limit as long as a wire maintains its position between the electrodes.”

3. The authors write that a laser 32 mW/cm^2 is applied and for clarity it should be stated that this is the light intensity that is incident on the nanowire.

Thanks for the comments. We changed the description accordingly to “A 532-nm laser at an intensity of 32 mW cm^{-2} is used to illuminate nanowires together with the dim LED to obtain the laser-induced alignment behaviors in the red curve.”

4. To make the change in alignment (from parallel to perpendicular) clearer, the authors should perhaps include a dashed line that guides the eye at $y=0$.

We added a dashed line for guidance in Fig. 2a,b.

5. On page 13 and in Fig 2: Strictly, I would think that you do not determine the real part of the polarizability, but the polarizability “anisotropy”.

We agree that the $\text{Re}(\alpha_{\parallel} - \alpha_{\perp})$ can be more precisely described by the term ---anisotropy of the real part of polarizability. We updated the manuscript accordingly.

6. At a given “voltage” p13; 240 should be electric field strength.

Thanks, we corrected it.

7. On page 18 it would be more general to also state the electric field strengths, not just the applied voltages (which depend on the electrode geometry).

We added another x-axis on the top of Fig. 4c to show the strength of the electric fields.

8. The text would still benefit from some more editing, e.g. here are some examples:

2;39: „environmental remedy“ should probably be remediation
4;68: broadly impacts
5;101 is “lithographed” an expression?
7;132 As soon as an electric

We thank the reviewer again. We carefully corrected the above errors and proofread another round.

Reviewer #2 (Remarks to the Author):

The authors have clarified the novelty of the manuscript when compared to their previous work (Science advances, 2018, 4(9), p.eaau0981) and addressed the inquiries raised from the reviewers by complementing the manuscript with some missing details. They also improved the quality of their manuscript by upgrading both figures and videos.

Therefore, I recommend the current manuscript for publication.

We sincerely thank the reviewer. We appreciate the efforts, valuable comments, and the support of the acceptance of our revised manuscript.

As a minor comment, In figure 3B, the σ legends overlap with the graph line.

We repositioned the legends in the updated Fig. 3B.